# Biotic resistance predictably shifts microbial invasion regimes

**Xiaozhou Ye ⓘ, Or Shalev & Christoph Ratzke ⓘ ✉**

Invading new territory is a central aspect of the microbial lifestyle. However, invading microbes rarely find novel territories uninhabited; resident microbes can interact with the newcomers and, in many cases, impede their invasion – an effect known as 'biotic resistance'. Accordingly, invasions are shaped by the interplay between dispersal and resistance. However, these two factors are difficult to disentangle or manipulate in natural systems, making their interplay challenging to understand. To address this challenge, we track microbial invasions in the lab over space and time – first in a model system of two interacting microbes, then in a multi-strain system involving a pathogen invading resident communities. In the presence of biotic resistance, we observe three qualitatively different invasion regimes: 'consistent', 'pulsed', and 'pinned', where, in the third regime, strong biotic resistance stalls the invasion entirely despite ongoing invader dispersal. These rich invasion dynamics could be qualitatively predicted with a simple, parameter-free framework that ignores individual species interactions, even for rather complex communities. Moreover, we show that this simple framework could accurately predict simulated invasions from different mechanistic models, indicating its broad applicability. Our work offers an understanding of how biotic resistance impacts invasions and introduces a predictive tool to identify invasion-resistant communities.

Due to their small size, microbes are true masters of dispersal, easily travelling with wind[1,2], water currents[3], or other organisms[4]. This dispersal ability allows microbes to reach even the most remote habitats[5]. The dispersing microbes only thrive under the right nutrients[6], pH[7], temperature[8], or salinity conditions[9]. Accordingly, abiotic environmental factors of the novel territory can decide about the establishment of an invading species. However, sequencing data of environmental samples show that microbe species are absent from many habitats that they in principle could live in[10,11]. Are even microbial invasions restricted by limited dispersal, or can other mechanisms also hinder microbial invasion? While dispersal rate (also known as propagule pressure) has been recognized as a key driver for macro-invasions[12–16], its importance in microbial invasion is inconsistent across studies and therefore remains unclear[17–22]. On the

other hand, microbial invasions can also be limited because novel territories are often already occupied by other microbes. These resident microbes can inhibit the establishment of invaders - an effect known as 'biotic resistance' or 'colonization resistance'[23–26] - and therefore potentially hinder invasions. The interplay between dispersal and the resident community's biotic resistance, therefore, strongly impacts the spatio-temporal dynamics of microbial invasion (Fig. 1a).

The impact of biotic resistance on invasions can also be relevant for us humans when the invaders are either crop[27,28] or human[29,30] pathogens. It is hypothesized that the resident microbiota in our body can inhibit invading pathogens and, in this way, protect us from diseases. In agreement, several studies have shown that beneficial gut microbiota can lower the risk of *C. difficile*[31,32] and enterobacteria[33–37]

Interfaculty Institute for Microbiology and Infection Medicine Tübingen (IMIT), Cluster of Excellence EXC 2124 "Controlling Microbes to Fight Infections" (CMFI), University of Tübingen, Tübingen, Germany. ✉e-mail: christoph.ratzke@uni-tuebingen.de

infections. Similarly, in plants, microbial communities of leaves[38–40] and roots[41–43] can suppress the growth of pathogens, hence protecting the plants from diseases. Microbial communities that impact diseases within a host may consequently also counteract the spread of infections between hosts. The potential importance of microbial biotic resistance for our personal and global health requires a better and ideally predictive understanding of this phenomenon.

To understand how biotic resistance and dispersal together shape microbial invasions, we must study ecological systems where we can independently manipulate both factors and observe invasions over space and time. In this work, we built microbial systems with different and partially tunable levels of biotic resistance. We used these systems to track invasions in real-time under various dispersal rates and observed that the interplay between dispersal and microbial interactions gives rise to different types of invasion dynamics. The obtained data allowed us to develop a parameter-free framework to predict under which conditions an invasion will be successful. This framework can be used to estimate the invasion resistance of microbial communities, which could, for example, be useful in identifying or even designing pathogen-resistant microbiota.

## Results

### Building microbial systems with varying biotic resistances

To study the impact of biotic resistance on microbial invasions, we first constructed microbial model systems with varying biotic resistances.

As a simple two-species system, we chose *Sporosarcina ureae* (Su) as an invader and *Lactiplantibacillus plantarum* (Lp) as the resident species. In this system, Lp inhibits the growth of Su by lowering the pH of the media (Supplementary Fig. 1 and previous study[44]), while adding buffer to the media weakens the ability of Lp to change the media's pH (Supplementary Fig. 1a). Accordingly, the more buffer we add to the media, the less Lp inhibits Su (Fig. 1b, c), weakening the biotic resistance of Lp against Su invasion. In this simple microbial system, the biotic resistance can thus be tuned by the buffer concentration of the media.

To obtain insights into more complex and realistic scenarios of invasion, we also studied the invasion of the pathogen *Pseudomonas aeruginosa* (Pa) into synthetic microbial communities comprising strains obtained from the *Caenorhabditis elegans* gut (the 'multi-strain system'). For that purpose, we built several resident communities by mixing various sets of strains and culturing them until the strain compositions reached equilibrium (Supplementary Fig. 2b, c). The obtained communities are composed of different species and differ in their biotic resistances, as can be seen in Fig. 1b, d. These different resistance levels cannot be explained by pH changes alone (Supplementary Fig. 2c). Different from previous studies[45], the community productivity (OD$_{600nm}$, Supplementary Fig. 2d) also does not determine biotic resistance in our systems. Interestingly, the most resistant community (Comm D) consists of *Pseudomonas* strains (Supplementary Fig. 2b), i.e., closely related to the invader, suggesting that niche similarity might play a role in determining resistance levels.

The obtained experimental systems offered us a unique opportunity to experimentally investigate the impact of biotic resistance on microbial invasions, as described below.

### Interplay between dispersal and biotic resistance causes three types of invasion dynamics

To observe the impact of biotic resistance on invasion dynamics, we conducted invasion experiments using our two- and multi-strain systems with varying biotic resistances. We performed the experiments along the rows of multiwell plates following a setup used in previous studies[46,47], filling the first 4 wells with the invading species and the subsequent 8 wells with the resident community. Every day we carried out dispersal by transferring a certain fraction (m) from each well to the neighboring wells. Upon dispersal, we also diluted the cultures into a new multiwell plate with fresh media and incubated them for 24 h to allow for species interaction (Fig. 2a, "Methods" section). We repeated this process for 10 days. We estimated the abundance of the invading species daily by measuring OD$_{600}$ or bioluminescence ("Methods" section) which correlate with invader fraction for the two- and multi-strain systems, respectively (Supplementary Figs. 3 and 4). The obtained data allowed us to follow the invasion in space and time and calculate how much distance (wells) the invasion front moved forward both on the n-th day (daily invasion speed $v_n$) and on average (mean invasion speed v) (Fig. 2b and "Methods" section). Leveraging this system, we conducted experimental invasions in our simple two-strain system as well as our more complex multi-strain system at varying dispersal rates and biotic resistances. The obtained results are shown in Fig. 2c–e.

We observed three distinct types of invasions: consistent (I), pulsed (II), and pinned (III) (Fig. 2b). In consistent invasion, the invasion front consistently moves forward without a stop (Fig. 2c, I). In pulsed invasion, the invasion front advances in bursts (fast invasion phases) separated by seemingly stationary periods (slow invasion phases) (Fig. 2c, II). During the slow invasion phase, the small invader population faces relatively strong biotic resistance and accumulates slowly at the invasion front (Supplementary Fig. 5e–g); during the fast invasion phase, the invader population growth is large enough to break through the biotic resistance and rapidly take over a new habitat. In pinned invasion, the invasion front is frozen in space and time (Fig. 2c, III). The invader can never overcome the biotic resistance of the resident population, despite ongoing dispersal and consistent arrival of invader cells to the resident community. Accordingly, the invader is permanently present in the invaded habitat patch but can never fully establish growth there, even over longer times (Supplementary Fig. 5a–d).

### At strong biotic resistance, invaders must overcome a critical dispersal rate to successfully invade

To understand further what determines the type of invasion dynamics, we examined invasions across a wide range of dispersal rates and biotic resistance levels. The resulting phase diagrams are shown for one representative replicate in Fig. 2d, e (for all replicates see Supplementary Figs. 6, 7). As can be seen for both the two- and multi-strain systems, consistent invasion can only happen for high dispersal rate and low biotic resistance. Upon increasing biotic resistance or lowering dispersal rate, the consistent invasion turns into a pulsed invasion, where the frequency of the pulses gradually decreases and the amplitude increases. Finally, for strong biotic resistance, complete pinning can occur if the dispersal rate falls below a critical value. In other words, resident communities with sufficient biotic resistance can withstand a certain inflow of invaders if it stays below this critical dispersal rate.

Interestingly, this rather simple phase diagram with three distinct regimes of invasion dynamics cannot only be observed for our two-strain system, but also for the invasion of *P. aeruginosa* into complex communities, where multiple species interactions determine the outcome. Observing such a simple pattern in such complex systems raises the question if the invasion dynamics could be predicted from simple features of the communities. Such a prediction would allow us to foresee the success of a possible invasion before it happens and identify microbial communities that are particularly resistant towards invasions, which would be of great use in a wide range of fields that deal with microbial invasions.

### Simple interaction measurements predict experimental invasion dynamics

Since an invasion is a combined outcome of dispersal and interaction, understanding both processes well enough should, in principle,

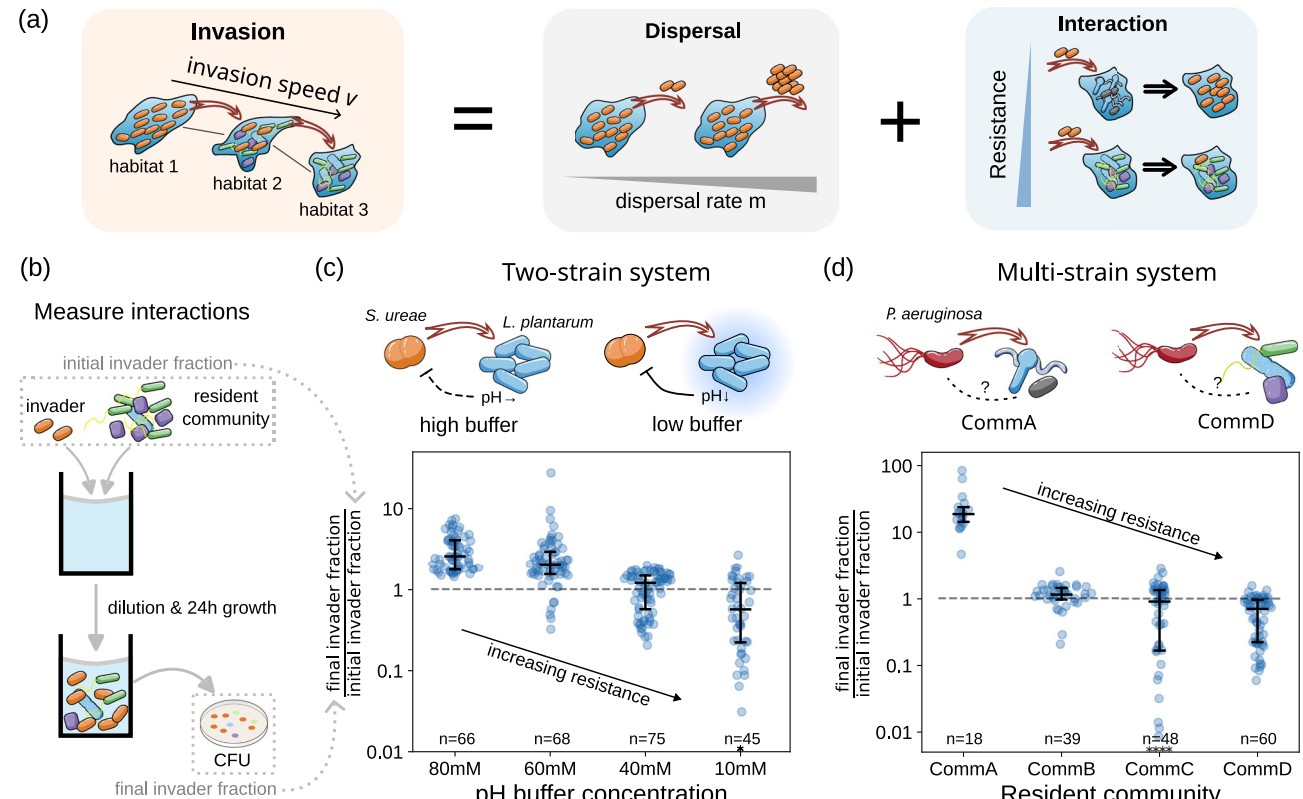

**Fig. 1 | Studying invasions as the combined outcome of dispersal and resistance. a** The dynamics of an invasion (left) depend on the dispersal rate (middle) and the resistance of the resident community (right). **b** We measured the interactions between the invader and the resident community to quantify the biotic resistance. We mixed different amounts of an invader to a resident population or community and measured the change of this invader's fraction after 24 h by colony forming units (CFU, see "Methods" section). A lower ratio of final versus initial invader fraction indicates a stronger biotic resistance. **c** In a two-strain system with *Sporosarcina ureae* (Su) as invader and *Lactiplantibacillus plantarum* (Lp) as resident species, the biotic resistance of Lp is mainly mediated by pH and can be tuned by changing the media's buffer concentration. **d** In a multi-strain system, four stably coexisting communities of *C. elegans* gut strains (CommA-D) show different biotic resistance levels toward the invading pathogen *Pseudomonas aeruginosa* (Pa). For (**c** and **d**), data from 3 biological replicates and different initial invader fractions are plotted if the final invader fraction is <0.9, and are scattered to reflect probability density distributions. Black horizontal bars denote median and quantiles, and black asterisks denote data points that lie outside the plotted areas. Source data are provided in Supplementary Data 1.

allow us to predict the invasion dynamics. However, in practice, we would have to know all interactions within the resident community as well as between the invader and resident species. Such information is, in general, not available. Thus, instead of trying to understand all the mechanistic details of the interactions, we coarse-grained the situation and treated the resident community as "one species" that interacts with the invader. This is quite a strong simplification since most microbial interactions in the system, as well as any change in the resident community's composition during invasion, are omitted. The interaction of the invader with the resident community can then be measured by mixing the invader and the resident community in different mixing ratios and measuring the invader fraction 24 h later. This shows us the impact of the whole community on the invader, a relationship we termed the interaction curve (Fig. 3a, left; "Methods" section). The interaction curve depends on both the interacting species and how these interactions are impacted by the abiotic environment, and thus summarizes both the biotic and abiotic resistance.

Next, we can use the obtained interaction curve to make predictions about invasion dynamics as follows.

Given the invader fraction $x_k^n$ on day $n$ and well $k, k \in 1, 2, \ldots K$, the updated invader fraction after dispersal (Fig. 3a, right) is given by

$$x_{ini\,k}^{n+1} = x_k^n \cdot (1-m) + x_{k-1}^n \cdot \frac{m}{2} + x_{k+1}^n \cdot \frac{m}{2} \tag{1}$$

After this simulated dispersal, the change of the invader fraction by species interactions is approximated by the interaction curve as

$$x_k^{n+1} = g\left(x_{ini\,k}^{n+1}\right) \tag{2}$$

(Figure 3a, left). Equations 1 and 2 can be combined into a single equation as shown in Fig. 3a, middle. In this way, we obtain a model that does not contain any free parameters and does not explicitly take into account the detailed species interactions in the system. We iterate the above procedure to generate predicted invasion dynamics over days, from dispersal rates and measured interaction curves only.

We next tested whether the proposed prediction framework could predict experimental invasions. To start, we measured the interaction curves as described in Fig. 3a, for our two-strain (Fig. 3b(i), Supplementary Fig. 8) and multi-strain systems (Fig. 3c(i), Supplementary Fig. 9). Then, we inserted the interaction curves into the prediction framework in Fig. 3a to predict invasion dynamics. It is important to note that we measured the interaction curves in a separate experiment and did not obtain them from the invasion data. Furthermore, the points on the interaction curve are not correlated with each other as we measured them from independent measurements, i.e., different bacterial cultures. For the two-strain system, the predictions not only capture the qualitative shift of invasion dynamics as resistance and dispersal rate change but are even quantitatively accurate (Fig. 3b). For the multi-strain system, the predictions still qualitatively capture the invasion dynamics and its transition from

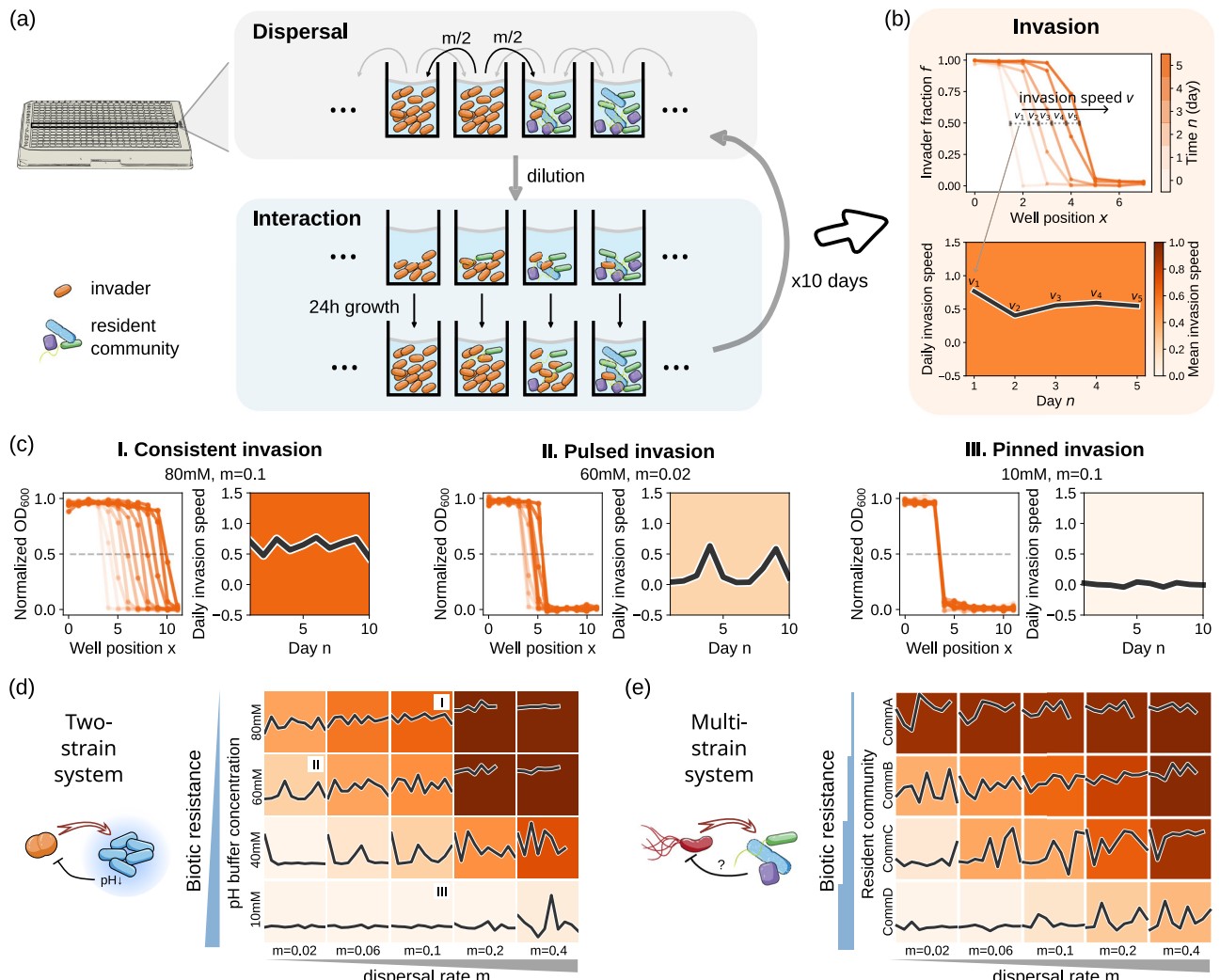

**Fig. 2 | Interplay between dispersal and biotic resistance gives rise to three distinct invasion dynamics. a** We performed invasion experiments in 384-well plates. We carried out dispersal by transferring a fraction m of the bacteria from each well to its neighboring wells. Afterwards, the invader and the resident community interact for 24 h. We repeated this process for 10 days, as described in more detail in the main text. **b** Repeated dispersal and interaction lead to an invasion wave (upper panel), from which the daily invasion speed (black line, lower panel) as well as the mean invasion speed over the duration of the experiment (background color, lower panel) can be obtained ("Methods" section). **c** Three qualitatively different types of invasions could be observed depending on dispersal rate and biotic resistance: consistent, pulsed, and pinned invasions. **d**, **e** Dispersal rate and biotic resistance determine the type of invasion in the two-strain (**d**) and the multi-strain system (**e**). The three examples in (**c**) are labeled by I, II, and III in (**d**). Source data are provided in Supplementary Data 1.

pinned to pulsed to consistent as dispersal rate increases and biotic resistance decreases (Fig. 3c). Our prediction however seems to underestimate invasion speed, possibly due to noise in experiments which has been theoretically shown to mostly accelerate invasion[48].

Interestingly, the observed interaction curves have rather complex shapes, showing that the residents' ability to resist the invader frequently varies depending on the relative abundance of the invader. Indeed, the overall shape of these interaction curves determines the invasion outcome more than the interaction at a fixed invader density, as discussed in more detail in the next section. The shape is especially important when the interaction curve crosses the diagonal. If the resident community can reduce invader fraction at low initial invader fractions but not at high initial invader fractions (e.g., 40 mM and 10 mM buffer in Fig. 3b; communities C and D in Fig. 3c) the dispersal rate becomes important for determining the invasion outcome, causing the emergence of a critical dispersal rate (Fig. 3b(iii), c(iii)).

The success of our simple prediction framework means that for a given invasion scenario, we could simply measure the interaction curve to get rather good answers to whether and how invasions would

proceed under different dispersal scenarios, without knowing the mechanistic details of the interactions. Measuring an interaction curve is much faster and easier than tracking an invasion over time, allowing us to predict invasions before they even occur. Moreover, since the interaction curve isolates the interaction component from the invasion process, it can be applied to different dispersal rates and invasion scenarios.

## Simulated invasions indicate that the prediction framework is widely applicable

To understand how generally the parameter-free framework of Fig. 3a can predict invasion dynamics, we used it to predict invasions simulated from two mechanistic models: the generalized consumer-resource model and the generalized Lotka-Volterra model. We first generated "ground truth" data with the mechanistic models as described in the "Methods" section and Fig. 4 captions, and then investigated how far the parameter-free framework can recapitulate these data (Fig. 4a, b). The invasion speeds predicted by the parameter-free framework correspond surprisingly well with the results of both

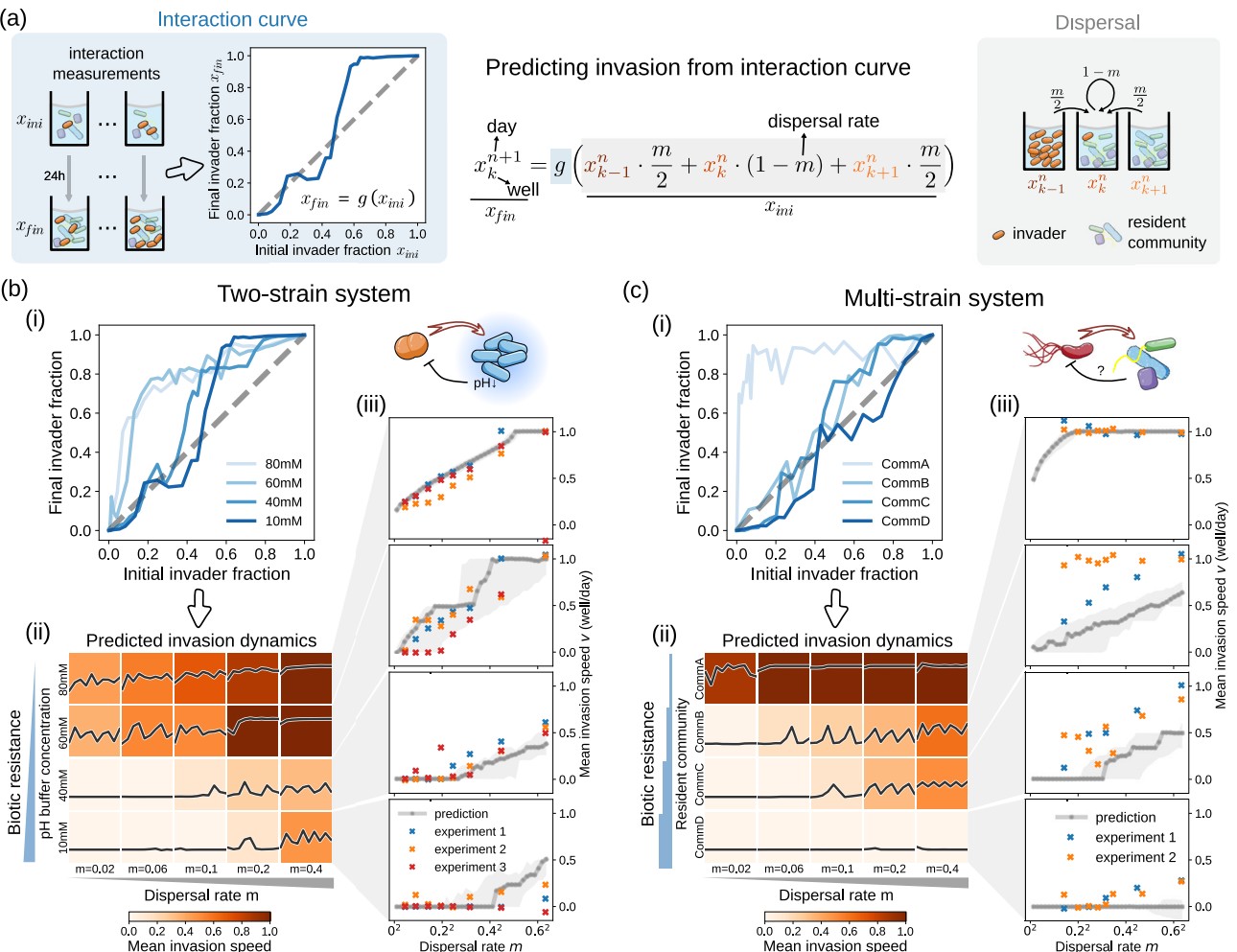

**Fig. 3 | A parameter-free framework predicts invasion in both two and multi-strain systems. a** The parameter-free framework leverages information about biotic resistance (blue background) and dispersal (grey background) to make predictions. The invader fraction in each well on each day is iteratively calculated from the values of the previous day, as outlined by the formula in the middle, which is a combination of Eqs. 1 and 2 of the main text. We approximate the biotic resistance with an interaction curve g(), which describes the change of the invader fraction after interacting with the resident community. The interaction curve is obtained by mixing the invader and the resident community in different mixing ratios and measuring the invader fraction 24 h later (**a**, left; also shown in Fig. 1b; "Methods" section). Dispersal is represented by a weighted sum of invader fractions in each well and its neighbors (**a**, right). Since the dispersal rate m is known for each experiment, there are no free parameters left for the model. Details of the

model are described in the main text. **b, c** Predicted invasion dynamics for two-strain (**b**) and multi-strain (**c**) systems. (i): Experimentally measured interaction curves for different biotic resistance levels, the mean of 3 replicates is shown. (ii) invasion phase diagram predicted from the measured interaction curves using the parameter-free framework depicted in (**a**); subpanels are as described in Fig. 2, x-axis is day n with range [1, 10], y-axis is daily dispersal rate $v_n$ with range [−0.5, 1.5]. (iii): predicted and measured invasion speeds as dispersal rate changes, for each biotic resistance level. Predictions from the averaged interaction curve (dark grey line) as well as from curves of mean interaction ±1 standard deviation (light grey shadow) are shown. The dispersal rate m is plotted on a square root scale, which is theoretically predicted to be proportional to invasion speed v in the absence of resistance[56]. Source data are provided in Supplementary Data 1.

mechanistic models, even for rather biodiverse resident communities (Supplementary Fig. 10b, e). In most cases, the invasion speed fluctuations are also predicted correctly (Supplementary Fig. 10c, f). Prediction failure happens only rarely and often for more complex invasion dynamics like backwards invasion and bidirectional invasion (Supplementary Fig. 12). The framework can also predict invasion in other dispersal scenarios, for instance, when dispersal is unidirectional instead of bidirectional (Supplementary Fig. 13).

Such high prediction accuracy is surprising, because contrary to the mechanistic models, our parameter-free framework neglects most microbial interactions as discussed above. Moreover, the interaction curve that describes the impact of the resident community on the invader is assumed constant and thus unaffected by the invader (Fig. 3). But in reality, we expect that the invader usually alters the composition of the resident community and thus in return how the community impacts the invader. We suspect that the parameter-free

framework is robust against this strong assumption because the success of an invasion is mostly decided in its early phases, where the density of the invader is low and its impact on the resident community is only minor. This statement is underlined by a close-up comparison of the predicted and simulated invasion dynamics that are discussed in more detail in Supplementary Fig. 11.

We further found that the predictions of our parameter-free model depend mostly on the shape of the interaction curve and less on the interaction outcome for a specific initial invader fraction. As a demonstration, Fig. 4c shows three interaction curves taken from the consumer-resource model, each reaching a similar final invader fraction upon starting with 1% invader (Fig. 4c, left, bottom), whereas the overall shapes of the curves are different (Fig. 4c, left, top). Despite their similarity at small initial invader fractions, the different curve shapes result in different invasion dynamics (Fig. 4c, right). Therefore, it may not be sufficient to study invasions at a fixed invader fraction.

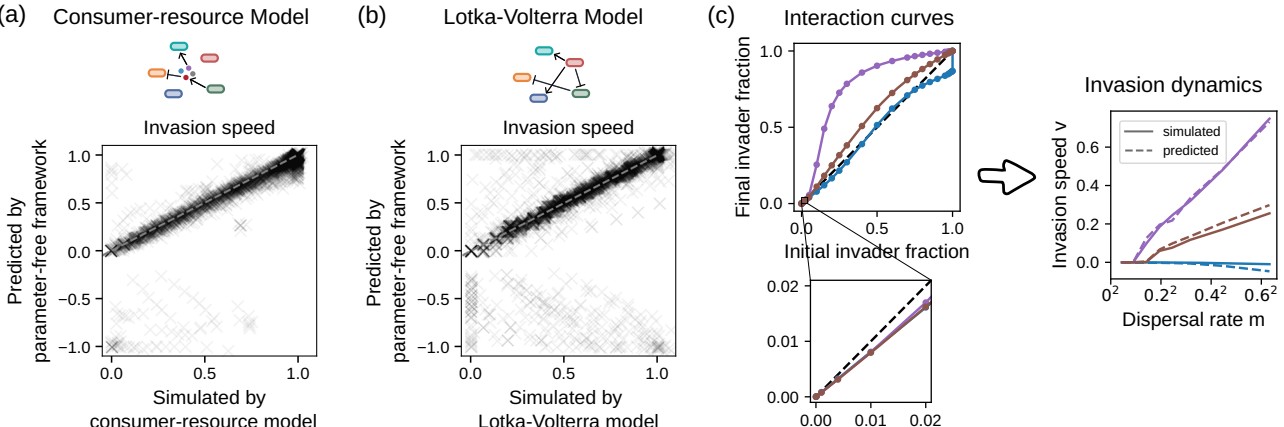

**Fig. 4 | Simulated communities indicate that the parameter-free prediction framework is widely applicable. a, b** Predicted vs. simulated invasion speed for invasion simulated by (**a**) generalized consumer-resource model where species interactions are mediated by biochemicals ($n = 16,000$, prediction error: median = 0.0071, 90% upper limit = 0.066); and **b** generalized Lotka-Volterra model where species directly impact the growth of each other ($n = 16,000$, prediction error: median = 0.0059, 90% upper limit = 0.12). For each model, we generated 100 resident communities (each with 4–15 stably coexisting species, Supplementary Fig. 10a, d) and 20 invader species with randomly drawn growth and interaction parameters. For each pair of invader and resident communities, interaction curves

were obtained from the mechanistic models and inserted into the parameter-free framework depicted in Fig. 3a to make predictions. The invasion simulations were performed equivalent to the experiments shown in Fig. 2a across 8 dispersal rates. Only scenarios with non-negative invasion speed are shown; other scenarios are discussed in Supplementary Figs. 11–14. **c** Interaction curves that appear similar at single measurements can be vastly different in overall shape (left), leading to different invasion dynamics (right). Each color corresponds to one invader-resident community pair from the consumer-resource model. The example pairs are chosen such that when the initial invader fraction is 0.01, the final invader fraction falls between [0.0079, 0.0081].

We provide a more intuitive understanding of the connection between the shape of the interaction curve and the resulting invasion process in the supplement by representative examples (Supplementary Fig. 14) and by a graphical approach called cobwebbing (Supplementary Fig. 15). Importantly, approximating the interaction curve with 3–5 data points is normally sufficient for accurate predictions (Supplementary Fig. 16). This high data-efficiency makes our parameter-free framework applicable even for invasions with limited data available. Overall, our results suggest that interaction curves summarize the key aspects of biotic resistance and can be used to predict invasion in a wide range of practical contexts.

## Discussion

Microbes usually intrude into space that is already occupied by resident organisms. Consequently, the invasion outcome not only depends on how fast an invader can spread into a novel territory and the abiotic conditions of the new habitat, but also on how it interacts with the resident organisms at the new location. Many current studies focus on one of the two aspects: either the spatial expansion of microbes into empty space[47,49,50] or the interactions between microbial invaders and residents without spatial context[22,51–55]. Our study extends these works by showing that complex yet predictable dynamics of invasions can emerge from the interplay between both dispersal and biotic resistance, as caused by species interactions.

Upon changing dispersal and biotic resistance, we observed three regimes of invasion dynamics in our experiments: consistent, pulsed, or pinned invasion waves. At weak biotic resistance, consistent invasion occurred, just like a classical invasion into empty space[56]. As biotic resistance increased and the dispersal rate decreased, pulsed and pinned invasion waves emerged. These three invasion regimes have been previously described by theoretical studies that modeled invasion as a single-species reaction-diffusion process. These studies identified two conditions for pulsed and pinned waves. The first condition is patchy instead of continuous habitats[57–59], as is the case for our experiment and many natural scenarios[60]. The second condition is the Allee effect[57,61–63], which means the growth rate of a species reduces when the population size drops, often due to the need for cooperative growth[64]. In our study, biotic resistance of resident communities

makes it more challenging for the invader to grow at low population density as compared to high density, leading to similar dynamics as caused by an Allee effect. Although we performed dispersal as a discrete daily event, pinned and pulsed invasion can also happen under continuous dispersal[57,58]. Our results indicate that simplified theories about the spread of a single species can be applied to more complex scenarios where resident communities are involved.

The occurrence of pulsed and pinned invasion can have important ecological consequences. During a pulsed invasion, the invasion seemingly comes to a stop at times, just to accelerate later again. Such dynamics could be misinterpreted as a halt to invasion when the observation period is too short. Pulsed invasion dynamics have been observed in macro-ecosystems[65] and were explained by stratified dispersal and the Allee effect of the invader. Our findings indicate that such dynamics can also be related to biotic resistance. Since natural habitats are usually already inhabited, we expect that biotic resistance to be a very common effect in nature. Pulsed waves are only mildly affected by spatial heterogeneity in dispersal rate (Supplementary Fig. 17) and carrying capacity (Supplementary Fig. 18); heterogeneity in growth rates has a bigger impact and can frequently lead to irregular pulses (Supplementary Fig. 19). At strong biotic resistance and low dispersal, pulsed invasion turns into pinned invasion, in which the invasion front is completely frozen in space and time. Despite ongoing dispersal, the pinned wave forms a stable boundary between the invader and the residents due to microbial interactions. Such boundary formation may explain some macroscopic patterns, such as the alternating distribution of savanna and forest[66].

Biological invasions can cause major damage to ecosystems and human society[29,30,67–70]. Accordingly, strong efforts are undertaken to stop invading species around the world. We show here that biotic resistance is crucial to support these efforts. Strong biotic resistance makes pinned and, therefore, a complete stop of an invasion possible, if the dispersal rate is below a critical value (i.e., critical dispersal rate). Without a critical dispersal rate, the invasion will progress even with a very low dispersal rate. Consequently, the invader must be completely obliterated to stop an invasion, which might be very challenging in practice, making biotic resistance almost a precondition for stopping an invasion.

Identifying a potential risk of invasion before its actual onset is a big challenge in ecology. Several approaches have been proposed to spot invasion risk based on traits of potential invaders[71–75] or properties of resident communities, such as biodiversity[22,55,76–81], but with mixed success. Our parameter-free framework takes an alternative approach; instead of focusing on specific properties of potential invaders and resident communities, it bases predictions on coarse-grained measurements of the interaction between the two (i.e., interaction curve). For a given invader and a corresponding resident community, the framework predicts how fast invasion would proceed under different dispersal rates, particularly if a critical dispersal rate exists. This framework allows us to identify invaders that would easily invade a given ecosystem and to estimate how much a given community can resist an invasion.

Our approach is, in principle, applicable across various ecosystems, although there are some limitations and precautions. For instance, measuring the biotic resistance and dispersal rate can be difficult for many macroscopic ecosystems. Also, for optimal prediction accuracy, the species interactions should be measured at a similar or shorter time scale than the occurrence of the dispersal events (see Supplementary Note 2 for clarification), which can become experimentally challenging in case of frequent dispersals. Nevertheless, such measurements are generally feasible in experimentally accessible microbial ecosystems. Hence, our approach may be applied to estimate the risk of disease spreading in the human gut or skin for a given microbiome profile.

Having a reliable indicator for biotic resistance can also be useful for manipulating, or even rationally designing microbial communities towards higher resistance. Many studies aim to understand how environmental factors impact the resistance of microbiota, such as drugs[82,83] and diet[84,85] in the human gut or light conditions in plants[86]. Moreover, teams from both academia[87,88] and industry[89–91] are trying to build resistant microbial communities from scratch. For both applications, it is essential to estimate the pathogen resistance of microbial communities. This is normally done by introducing a certain amount of pathogen into the microbiota and observing its growth, which may not capture all features of biotic resistance as we discussed in Fig. 4c. We extend this approach by using slightly more measurements to obtain the interaction curve and, through the prediction framework, linking it directly to predicted invasion outcomes. Importantly, the framework predicts invasion for any given dispersal rate, thus providing a more comprehensive definition of biotic resistance.

The other way around, microbiota engineering can also require a successful invasion. For instance, gut microbiota are transferred from healthy to sick people (i.e., fecal transplantation)[32,92,93] to treat gut-related diseases, or specific bacteria strains are consumed (i.e., probiotics) to restore healthy gut microbiota[94–98]. Similarly, there are attempts to fight plant pathogens by introducing protective bacteria[99–101], or to improve productivity by adding fertilizing microbes[102,103]. In all cases, the introduced microbes have to invade the resident microbiota to fulfill their task, which can often be challenging[104,105]. Our framework could predict invasion outcomes for different doses and frequencies of microbe treatment (i.e., different dispersal) and help optimize therapies. Specifically, if a critical dispersal rate is predicted, a sufficient number of microbes must be added to overcome it.

Taken together, we have shown that biological invasion into occupied territory results in a rather simple set of invasion dynamics, despite complex species interactions and spatial processes. These dynamics can moreover be predicted by a straightforward, parameter-free framework, suggesting that it is feasible to understand and manipulate invasion for our benefit.

## Methods
### Preparation of media and agar plates
Bacteria were precultured in nutrient media (NM), consisting of 1% yeast extract (Sigma–Aldrich #70161) and 1% soytone (Peptone from soybean, Sigma–Aldrich #87972). Both interaction and invasion experiments were performed in invasion media (IM), which resembles diluted De Man, Rogosa, and Sharpe Media (MRS) with some compounds substituted by more commonly used chemicals in our lab. IM consists of 0.1% soytone, 0.1% tryptone (Sigma–Aldrich #95039), 0.04% yeast extract, 7.34 mM ammonia acetate (Merck #1.01115.1000), 1.64 mM tri-sodium citrate (VWR #27833.237), 0.8 mM $MgSO_4$, 0.2 mM $MnSO_4$, 0.4% glucose, 12 mM $(NH_4)Cl$, 12 mM NaCl. Different concentrations of phosphate buffer (pH = 6.8, made from 0.58 M $K_2HPO_4$ and 0.42 M $KH_2PO_4$, followed by pH adjustment) were added to IM, as specified for each condition in the two-strain system and fixed to 10 mM for the multi-strain system. Before each experiment, IM was freshly assembled from 6 stocks with partial components (Supplementary Table 1), sterilized by a 0.2 µm filter (Filtropur S, Sarstedt #83.1826.001), and stored at 4 °C for up to 2 weeks. Such assembly from partial stocks reduced interactions between components during storage and enhanced reproducibility.

Colony-forming units (CFU) of bacterial cultures were quantified on different varieties of NM agar plates depending on the experiment. NM agar plates of pH 5 and pH 9 were prepared from 1% yeast extract, 1% soytone, 10 mM $NaH_2PO_4$, and 2.5% agar (Agar–Agar, Carl Roth #5210.2), pH was adjusted before autoclave sterilization. pH7 NM agar plates with/without gentamicin were prepared from 1% yeast extract, 1% soytone, and 2.5% agar, for plates with antibiotics, 5 µg/mL gentamicin (Gentamicin sulfate, Acros Organics #455310050) was added after the agar partially cooled down following autoclave sterilization. 45 mL autoclaved agar mixture was poured into a 150 mm petri dish for each agar plate.

### Bacterial strains
In the two-strain system, Sporosarcina ureae (Su) was purchased from Ward's Science (#470179-156), and Lactiplantibacillus plantarum (Lp) from ATCC8014. In the multi-strain system, the invader *Pseudomonas aeruginosa* (Pa) is a PA01 strain, which we genetically tagged with luciferase (see below). The 16 strains that we used to build resident communities (Supplementary Fig. 2) were previously isolated from the gut of *Caenorhabditis elegans* (C. elegans) from Northern Germany[106] (MYb27) or Massachusetts, United States[107] (the other 15 strains). The 16S sequence of the V3–V4 region for these 16 strains is available in Figshare (https://doi.org/10.6084/m9.figshare.27168834).

**Luciferase tagging of the invader in the multi-strain system.** *Pseudomonas aeruginosa PA01* ('PA01') was genomically transformed with the luxCDABE operon using the mini-Tn7 insertion system as previously described[108]. Briefly, the plasmids pUC18T-mini-Tn7T-Gm-lux (Addgene #64953) and pTNS2 (Addgene #64968) were extracted from E. coli DH5a using a plasmid extraction kit (peqGOLD Plasmid MiniPrep Kit II; VWR, Leuven, Belgium). PA01 cells were made competent for transformation by treatment with a 300 mM sucrose solution, and were then electroporated with 50 ng of the isolated pUC18T-mini-Tn7T-Gm-lux and pTNS2 plasmids. After recovery in 1 ml of LB at 30 °C, the electroporated cells were plated on an LB-agar+Gm30 plate. Plates were incubated for 24 h at 30 °C, and colonies were tested for their bioluminescence activity using a plate reader. Positive colonies were then further grown in a selective medium (LB+Gm30) and were stocked in 25% glycerol at −80 °C.

### $OD_{600}$ and bioluminescence measurement
**$OD_{600}$ (two-strain system).** A 384-well plate with bacterial cultures was mixed for 15 s at 2000 rpm on a Mixmate Plate Shaker ('Mixmate', Eppendorf, Hamburg, hereon shortened as Mixmate), then put into the FLUOstar Omega Plate Reader (BMG Labtech, hereon shortened as plate reader) to measure absorbance at 600 nm.

**Bioluminescence (multi-strain system).** The bacterial cultures were transferred to a Lumox 384-well plate (Sarstedt 94.6130.384) for bioluminescence measurement. First, 2.5 μL NM was dispensed into the Lumox plate with Echo525 Liquid Handler (Beckman, hereon shortened as Echo525), then 20 μL bacteria culture was reverse dispensed into the Lumox plate using Viaflo384 (Integra) and mixed with the NM for 30 s at 1500 rpm in Mixmate. The Lumox plate was incubated at 30 °C in a plate reader for 2 min, allowing the added NM to reactivate potentially stagnated bacterial (Pa) metabolism, which gave rise to a more reliable bioluminescence signal. Finally, bioluminescence was measured with a gain of 3600, and exposure time 1 s on our FLUOstar Omega Plate Reader (BMG Labtech).

The raw bioluminescence measurement was corrected for signal crosstalk and background using an empirically fitted function. For the measured bioluminescence value $x_{ij}$ of row i and column j, the corrected value $\hat{x}_{ij} = x_{ij} - 0.087x_{(i+1)j} - 0.020x_{(i-1)j} - 0.058x_{i(j+1)} - 0.034x_{i(j-1)} - 20$. Before correction, signal crosstalk can lead to up to +1800 (a.u.) measurement error, which was reduced to within ±200 (a.u.) after correction.

## Community composition based on colony-forming unit (CFU)

In interaction experiments as well as representative invasion experiments, we obtained the species composition of bacterial cultures by CFU. First, we serially diluted cultures 1/10 in PBS until reaching a dilution factor of $1/10^6$, the dilutions were carried out in a plate-to-plate manner. Then, for each 96-well intermittent pattern in the 384-well plate, we used Viaflo384 to reverse pipette 5 μL from each dilution onto different varieties of NM agar plates, getting 96 separate droplets on the agar surface. For the two-strain system, we used pH 5 agar plates for selective plating of Lp and pH 9 agar plates for Su. For the multi-strain system, we used pH 7 agar plates with gentamicin for selective plating of Pa and pH 7 agar plates without gentamicin for both the resident community and Pa, on which colonies of Pa could be morphologically identified under a stereoscope. After the droplets dried out, we incubated the agar plates at 30 °C for 24 h until colonies grew to a distinguishable size. We then either took photos of the plates under ring light illumination and counted CFU afterward (for the two-strain system) or directly counted CFU under the stereoscope (for the multi-strain system).

## Community composition based on 16S sequencing

For the multi-strain system, the resident communities' compositions were confirmed by 16S sequencing (Supplementary Fig. 2b, c). We extracted DNA from stable cultures of resident communities using the peqGOLD Bacterial DNA Kit (VWR #13-3450-02), then sent the obtained DNA to Eurofins for library preparation and sequencing of the 16S rDNA V3–V4 region. To determine species composition, we aligned the obtained reads against a custom library comprising the V3–V4 region of the 16 species that we used, which were previously obtained by Sanger sequencing[107]. We filtered out chimera reads using vsearch (version 2.28.1) '–uchime_ref' option, then carried out alignment using bwa-mem with default parameters.

## Assembly of stable resident communities in the multi-strain system

**Summary.** As shown in Supplementary Fig. 2, we randomly assembled 384 communities each from a subset of a pool of 16 bacteria strains. We cultured the resulting community in 50 μL IM (10 mM buffer) on a 384-well Labcyte plate (Beckman #001-1455) for 8 days, with 1/200× daily dilution. The 384 initial communities collapsed into ~5 different representative compositions as indicated by CFU morphologies. We picked 4 wells with distinct compositions and stocked in 30% glycerol at −80 °C. The frozen stocks were revived and cultured for 4 additional days with a 1/200× daily dilution before being used for experiments. More details are as follows.

**Preculture and handling of single strains.** We assembled the resident communities used in the multi-strain system from 16 strains that were previously isolated from *C. elegans* gut[109]. The strains cover diverse phylogeny (Supplementary Fig. 2b) and have distinct colony morphologies on NM agar plates. We streaked out the 16 strains on pH 7 NM agar plates, then took a single colony from each strain and precultured in 200 μL TSB (Tryptic soy broth non-animal origin irradiated, VWR #84674.0500) in a 96-well 500 μL deep plate (Masterblock 96-well, Greiner #786201). We sealed the plate with 2 layers of Breathable Rayon Films (VWR #391-1261, hereon shortened as breathable films), then incubated on Titramax 1000 shaker (Heidolph) at 1350 rpm and 30 °C overnight (16 h). We then washed precultures twice and resuspended each in 200 μL IM (10 mM buffer).

**Assembly and daily dilution.** We transferred 50 μL of washed bacteria preculture to a 384-well Labcyte plate, as the source plate for community assembly with the Echo525 Liquid Handler (Beckman, hereon shortened as Echo525). We added 50 μL/well IM (10 mM buffer) to another 384-well Labcyte plate as the destination plate. For each well in the destination plate, we randomly assigned the presence ($p = 0.75$) or absence ($p = 0.25$) for each species, then programmed Echo525 to transfer from the source plate 25 nL of each species that was assigned presence for the current well. This resulted in 384 different initial communities consisting of 0–16 species (mean = 12), and 2 technical replicates with identical community layouts. We sealed the destination plate with 2 layers of breathable films, incubated on a Titramax 1000 shaker at 1350 rpm and 30 °C for 24 h. We continued culturing the communities for 7 consecutive days with 1/200× daily dilution, by using Echo525 to transfer 250 nL into a new 384-well Labcyte plate with 50 μL/well fresh IM (10 mM buffer).

**Freezing, reviving, and preparing glycerol stocks of the communities.** On day 8, we mixed 25 μL of each resulting community with 25 μL of 60% glycerol and froze the plate at −80 °C. To observe community compositions, we diluted the communities with a 1/10× dilution series and droplet plated 5 μL of the 1e-7, 1e-6, and 1e-5 dilutions using Viaflo384 onto a pH7 NM agar plate. The 384 communities collapsed into ~5 distinct compositions, as indicated by plated CFU morphologies (Supplementary Fig. 2b). We selected 4 communities with distinct compositions, thawed the frozen cultures, and cultivated them in 384-well Labcyte plate and 50 μL IM (10 mM buffer), with 1/200× daily dilutions for 8 more days to stabilize the communities. We froze the resulting cultures in 30% glycerol at −80 °C. We revived resident communities from these stocks by 1/200× daily dilution into a 384-well Labcyte plate with 50 μL IM (10 mM buffer) and culturing on a Titramax 1000 shaker at 1350 rpm and 30 °C in between. We confirmed by CFU and 16S sequencing that community compositions reached equilibrium within 3 days from the frozen stocks (Supplementary Fig. 2b). We performed all multi-strain system experiments from these stocks.

## Invasion experiments

**Overview.** We carried out an invasion for each given resistance level and dispersal rate m along 12 wells in a 384-well plate, mimicking 12 habitat patches. We cultured the invader and the resident community separately for 3 days to stabilize the potential initial dynamics, then initialized the invasion experiment with 4 wells of invader followed by 8 wells of resident community. Every day during the invasion experiment, we used the Echo525 Liquid Handler to perform dilution and dispersal from the current plate to a new plate with fresh media. Specifically, given the dilution rate $\delta$ (1/200× unless otherwise specified) and dispersal rate m (8 different values, as specified below), we diluted $\delta \cdot (1 - m)$ of the total volume from each well to the same well in the new plate, and $\delta \cdot \frac{m}{2}$ of the total volume to its left and right neighboring wells respectively (or the same well if no left/right

neighbor) in the new plate. Essentially, we performed a bidirectional dispersal between adjacent wells. After dispersal and dilution into fresh media, we cultured the new plate and let the species interact for 24 h (see below for study system-specific culture conditions). We recorded the spatial distribution of the invader daily by measuring OD600 for the two-strain system/bioluminescence for the multi-strain system, and for one experiment (Supplementary Fig. 5) also community composition by counting distinct colony-forming units (CFU). We repeated this dispersal-interaction cycle for 8–10 days. We performed 3 biological replicates for the two-strain system and 2 biological replicates for the multi-strain system, the replicates were run independently and in different months.

**Two-strain system.** We streaked out Su and Lp from glycerol stocks to pH 7 NM agar plates and grew them at 30 °C for a day. We picked a single colony from each species with an incubation loop and cultured it in 5 mL NM in a falcon tube with the lid fully closed to avoid cross-contamination (the species still grew), on Innova 2000 shaker (New Brunswick, Eppendorf AG) at 225 rpm and 30 °C overnight (-16 h). We took 400 μL from the preculture for each species, washed the sample twice, and resuspended the cells in 400 μL IM (without buffer). We transferred 250 nL of the resulting cell suspension into 384-well Labcyte plates that contained 50 μL IM with corresponding concentrations of phosphate buffer, using Echo525. We sealed the plates with two layers of breathable films and incubated them on Mixmate at 1500 rpm and 30 °C for 24 h. We cultivated the bacteria for 3 more days with a 1/200× daily dilution to eliminate memory from the preculture and to stabilize the potential population dynamics before starting the invasion experiment. The daily dilution was performed by transferring 250 nL with the Echo525 from the previous plate to a new plate with 50 μL IM with corresponding buffer concentrations.

We performed invasion experiments in Labcyte 384-well plates with 50 μL IM with corresponding buffer concentrations. We used each one of the four intermittent 96-well patterns in the 384-well plate to perform invasion experiments for one buffer concentration. The invasion was carried out along each row (12 wells) of the 96-well pattern, every row corresponded to a different dispersal rate m ($m$ = [0.002, 0.008, 0.02, 0.038, 0.06, 0.1, 0.2, 0.4]), and was initiated by diluting 250nL of Su (first 4 wells) or Lp (last 8 wells) from the corresponding preculture into 50 μL IM with the corresponding buffer concentration, using Echo525. The experiments consisted of dispersal and 1/200× daily dilution, followed by 24 h growth on Mixmate at 1500 rpm and 30 °C. Every 24 h, we started the transfer by mixing the cultured plate for 15 s at 2000 rpm with the Mixmate, then used Echo525 to dilute 250 nL into another 384-well Labcyte plate with 25 μL fresh IM. From the original plate (1×) and the diluted plate (1/100×), we used the Echo525 to perform dispersal and dilution into a new 384-well Labcyte plate as specified in Supplementary Table 2. After the transfers, we sealed the plates with two layers of breathable films and incubated them in the Mixmate at 1500 rpm and 30 °C for 1 day (slightly less than 24 h, due to the time needed for performing dilution and dispersal). The original plate (1×) was used to measure OD$_{600}$, and in cases where community composition was needed, the diluted plate (1/100×) was serially diluted and plated for CFU analysis ("Methods" section).

**Multi-strain system.** The experiment followed the same principle as the Su invading into Lp experiment, with a few modifications due to the technical difficulties regarding reliably transferring Pa and the communities using Echo525, as specified below.

*Modified culturing media.* Instead of using IM with different phosphate buffer concentrations, the phosphate concentration was fixed to 10 mM.

*Modified preculture procedure.* We did the first preculture of Pa in 5 mL NM similarly to Su/Lp. We did the first preculture of the

community by 1/40 dilution from glycerol stock to 50 μL IM (10 mM buffer) in a 384-well Labcyte plate. We cultured the plate on a Heidolph Titramax 1000 shaker at 1350 rpm and 30 °C for 24 h. After the first preculture, we cultured both Pa and the communities in a 384-well Labcyte plate for 3 more days with a 1/200× daily dilution to stabilize possible initial community dynamics. We carried out the dilutions with the tip-based Viaflo384 pipetting system since the undiluted cultures of several communities incurred recurring transfer errors in the sound-based Echo525 system.

*Modified liquid transfer during the invasion.* Compared to the Su/Lp invasion experiments described above, we slightly modified dispersal rates ($m$ = [0.02, 0.04, 0.06, 0.08, 0.1, 0.12, 0.22, 0.4]) to use only 1/10× diluted bacterial culture in source plates. This modification was made because 1× bacterial community cultures contained lots of metabolic byproducts, which caused frequent Echo525 transfer failures, and in overdiluted Pa culture, like 1/100×, Pa tended to gather on the surface and impaired accurate Echo525 liquid transfer. Every day, we used Viaflo384 to dilute 5 μL of bacteria culture into a 384-well Labcyte plate with 45 μL IM (10 mM buffer). We mixed the 1/10× diluted plate 15 s at 2000 rpm with Mixmate, let it rest for 2 min, then performed dispersal and dilution with the Echo525 as specified in Supplementary Table 3. We used each one of the four intermittent 96-well patterns in a 384-well plate to perform invasion experiments with one resident community. After liquid transfer for dispersal and dilution, we cultured the plate on a Titramax 1000 shaker at 1350 rpm and 30 °C for slightly less than 24 h before performing the next dispersal.

**Invasion speed.** We defined the invasion front as the furthest well where the invader population crossed the 50% threshold, i.e., the midpoint of the invader's OD$_{600nm}$/bioluminescence values and the value of the resident community. We defined daily invasion speed on day $n$ ($v_n$) as the distance (in wells) the invasion front moved forward on that day, and calculated the mean invasion speed v as the average daily invasion speed from day 3 until the end of the experiment or until the invader took over all wells.

### Interaction experiments

**Overview.** Precultures were prepared following a similar procedure to that in invasion experiments. To get 3 biological replicates, 3 independent precultures were initiated from different colonies (for Su, Lp, and Pa) or independent aliquots from glycerol stock (for resident communities). We initiated the interaction experiments by mixing the invader and the resident cultures at different fractions and diluting 1/200 into a 384-well Labcyte plate with 50 μL IM (with corresponding buffer concentrations). The resulting interaction plate was cultured for 24 h to allow species interactions. The precultures were 1/10 serially diluted and plated to obtain initial per volume CFU. After 24 h, we measured OD$_{600}$ (two-strain system) or bioluminescence (multi-strain system) of the interaction plate, then 1/10 serially diluted the interaction plate and plated it onto corresponding plates to obtain final species composition by CFU ("Methods" section). Details are as follows.

**Two-strain system.** For each species, replicate, and buffer concentrations, precultures from day 3 were diluted 1/200× into 32 wells by using Echo525 to transfer 250 nL preculture into 50 μL IM with corresponding buffer concentrations. The resulting plate was sealed with double-breathable films, cultured on Mixmate at 1500 rpm and 30 °C for 24 h, and then used to initiate interaction experiments. We started by shaking the preculture plate (1×) for 15 s at 2000 rpm with the Mixmate, then diluted 500 nL into 49.5 μL fresh IM with corresponding buffer concentrations using Echo525. From the original plate (1×) and the diluted plate (1/100×), we used Echo525 to mix the invaders and the residents at 32 different invader ratios ranging from 0, 0.001 to 0.9, 1, and to dilute the mixture 1/200× into 384-well Labcyte plate with 50 μL IM (with corresponding buffer concentrations), as

detailed in Supplementary Table 4. After the transfers, we sealed the interaction plate with two layers of breathable films and incubated them in Mixmate at 1500 rpm and 30 °C for 24 h. 8 wells of precultures from each species, replicate, and buffer concentrations were 1/10× serial diluted and plated on pH 5 (for Lp) and pH 9 (for Su) NM agar plates to obtain CFU-based initial population size.

After 24 h, OD600nm and CFU-based species composition were measured for the entire interaction plate ("Methods" section). We carried out the serial dilution for CFU with Echo525, for each step, we diluted 2.5 μL of culture from the previous dilution into 22.5 μL PBS. The diluted plates were first mixed using Mixmate at 2000 rpm for 15 s, then, for each well, 5 μL of liquid was plated onto pH 5 (for Lp) or pH 9 (for Su) NM agar plates to obtain species composition by CFU ("Methods" section).

**Multi-strain system.** For each species/resident community and replicate, precultures from day 3 were diluted 1/200× into 32 wells with 50 μL IM (10 mM buffer) using Viaflo384. The resulting plate was sealed with double-breathable films, cultured on a Titramax 1000 shaker at 1350 rpm and 30 °C for 24 h, and then used to initiate interaction experiments. We started by shaking the preculture plate (1×) for 15 s at 2000 rpm with the Mixmate, then diluted 5 μL into 45 μL fresh IM (10 mM buffer) using Viaflo384. From the diluted plate (1/10×), we used Echo525 to mix the invaders and the residents at 25 different invader ratios ranging from 0.01 to 0.9 and to further dilute the mixture 1/20× (1/200× in total) into a 384-well Labcyte plate with 50 μL IM (10 mM buffer), as detailed in Supplementary Table 5. After the transfers, we sealed the interaction plate with two layers of breathable films and incubated it in a Titramax 1000 shaker at 1350 rpm and 30 °C for 24 h. 16 wells of precultures from each species/resident community and replicate were 1/10× serially diluted and plated on pH 7 with gentamicin (for Pa) and pH 7 without gentamycin (for community) NM agar plates to obtain CFU-based initial population or community size.

After 24 h, bioluminescence and CFU-based species composition were measured for the entire interaction plate ("Methods" section). We carried out the serial dilution for CFU with Viaflo384, for each step, we diluted 5 μL of culture from the previous dilution into 45 μL PBS. The serial diluted plates were mixed using Mixmate at 2000 rpm for 15 s, then 5 μL of liquid was plated onto pH 7 with gentamicin (for Pa) or pH 7 without gentamycin (for community) NM agar plates and used for obtaining species composition by CFU ("Methods" section).

**Obtaining interaction curves from interaction measurements**
To derive interaction curves from interaction measurements, the units for initial and final invader fractions should be the same. For our experimental setup, it is thus necessary to convert between CFU-based invader fraction $f_{CFU}$ and culture volume-based invader fraction $f_{VOL}$, as below. The initial invader fraction was first defined by volume as the fraction of invader preculture to be mixed with the resident preculture ($f_{VOL}$). Using measured CFU per volume data for the invader ($K_{ivd}$) and resident precultures ($K_{rsd}$), the CFU-based invader fraction could be calculated as $f_{CFU} = \frac{f_{VOL} \cdot K_{ivd}}{f_{VOL} \cdot K_{ivd} + (1 - f_{VOL}) \cdot K_{rsd}}$. The final invader fraction was first measured based on CFU ($f_{CFU}$), then the volume-based $f_{VOL}$ could be calculated by reversing the above relationship. Thus, interaction curves could be expressed either by CFU fraction or volume fraction (Supplementary Figs. 8 and 9) and could be converted to each other. Which of these two versions of the interaction curve to use for prediction depends on how the dispersal rate is defined. For instance, our invasion experiments defined dispersal rate as the volume of liquid transfer, so the volume-based interaction curves were used to predict invasion dynamics.

The functional form of interaction curves $g()$ used in predictions was obtained by linear interpolation of experimentally measured or

model-simulated initial and final invader fractions. The only exception was in Supplementary Fig. 15, where we inferred $g()$ from a small number of data points; there, we used Pchip interpolation[110] to fit the data with a smooth and monotonic curve.

**Statistics & reproducibility**
No statistical methods were used to predetermine sample size. All experiments were replicated at least 3 times. One replicate in the invasion experiment for the multi-strain system failed due to massive cross-contamination during the liquid handling process; because of its minor contribution in supporting the main message and the high cost and time requirement, no new replicate was added to compensate for the failed one. All other replicates were successful. A few points in interaction measurements or invasion experiments were excluded due to cross-contaminations or liquid transfer errors. The experiments were not randomized. Blinding was not necessary for the experiments: the invasion outcome was tracked by OD600nm, bioluminescence, or colony morphologies, all of which were objective measurements that would not change at the whim of the investigators.

**Generalized consumer-resource model**
In our generalized consumer-resource model, species interactions are mediated by biochemicals in the environment, which include both resources and toxins. Different species use, secrete, and are inhibited by different biochemicals, allowing all major types of bacterial interactions: resource competition, cross-feeding, inhibition by toxins, and cross-protection.

Denoting the population size of species i as $N_i$, the growth of species is described by:

$$\frac{dN_i}{dt} = r_i(\mathbf{R}) \cdot N_i \qquad (3)$$

in which $r_i(\mathbf{R}) = r_i^0 \cdot k_i(\mathbf{R})$ is the realized growth rate of species i, determined by the intrinsic growth rate of species i $r_i^0$ and the concentration vector of biochemicals $\mathbf{R}$ as follows:

$$k_i(\mathbf{R}) = \begin{cases} 0, & \sum_j c_{ij} R_j \geq 1 \\ \left( \frac{\sum_{j \in \{p_{ij} > 0\}} p_{ij} R_j}{R_M + \sum_{j \in \{p_{ij} > 0\}} p_{ij} R_j} \right) \left( 1 - \sum_j c_{ij} R_j \right), & \text{otherwise} \end{cases} \qquad (4)$$

For each biochemical j, species i has a preference value $p_{ij} \geq 0$ and is inhibited by a coefficient $c_{ij} \geq 0$. The majority of $p_{ij}$ and $c_{ij}$ are 0, meaning that species i doesn't consume biochemical j nor is inhibited by biochemical j, respectively. $R_M$ is the Michaelis-Menten coefficient used to capture the dependency of growth rate on available biochemicals that a species can consume (i.e., resources). Motivated by experimental observation that growth stops abruptly as resources run out, a small value of $R_M = 0.01$ was used. Biochemical j can also inhibit the growth of species i and reduce its growth rate proportional to $c_{ij} R_j$, until $r_i(\mathbf{R})$ is reduced to 0.

Species can both consume and secrete biochemicals. The biochemical concentration $R_j$ changes accordingly:

$$\frac{dR_j}{dt} = -\sum_i \hat{p}_{ij} \left( r_i^0 + \sum_k b_{ik} \right) k_i(\mathbf{R}) N_i + \sum_i b_{ij} k_i(\mathbf{R}) N_i \qquad (5)$$

Species i secretes biochemical j at a rate $b_{ij} k_i(\mathbf{R}) N_i$ during growth. We assume an energy density of 1 for all biochemicals, i.e., we normalize biochemical concentrations by their usable energy. The normalized resource usage $\hat{p}_{ij} = \frac{p_{ij} \cdot [R_j > 0]}{\sum_{k \in \{k : R_k > 0\}} p_{ik}}$, modelling a simplified scenario

where species consume the available resources simultaneously and proportionally to their species-specific preference to these resources. Species consume as many resources as the sum of their needs for biomass growth and secretion, so that secreting biochemicals increases resource uptake rather than reducing growth rate. This is to capture the observation that many secreted biochemicals are metabolic byproducts and would not incur a significant cost to bacteria[111].

### Generalized Lotka-Volterra model

In our generalized Lotka-Volterra model, each species j directly inhibits or facilitates species i, through negative and positive interaction coefficient $A_{ij}$ respectively. The growth of species i is described by

$$\frac{dN_i}{dt} = r_i(\mathbf{N})\left(1 + \sum_j A_{ij}N_j\right)N_i \tag{6}$$

$r_i(\mathbf{N}) = r_i^0 \left(\frac{K - \sum_i N_i}{K_m + (K - \sum_i N_i)}\right)$ is the realized growth rate of species i, where $r_i^0$ is the intrinsic growth rate of species i, $K = 3$ represents a relatively large carrying capacity, and $K_m = 0.1$ is a relatively small coefficient that sharply reduces the growth rate when the community approaches the carrying capacity. In this setting, $r_i(\mathbf{N}) \approx r_i^0$ in most cases, however, growth is constrained by a carrying capacity shared by the community, preventing unbounded growth caused by positive interactions.

### Invasion, interaction curve, and prediction in mechanistic models

**Random generation of resident communities and invaders.** For both the generalized consumer-resource model (CR model) and the generalized Lotka-Volterra model (LV model), we generated 100 random resident communities (each starting with 50 species) and 20 random invaders. We studied the invasion dynamics for each of the 100*20 pairs. See Supplementary Note 1 for the choice and distribution of parameters.

We simulated resident communities from initial populations of $10^{-3}$ per species for 30 dilution cycles when most community compositions reached equilibrium. In each dilution-growth cycle, the species (and biochemicals for the CR model) were diluted by 1/200× from the previous cycle, as in the experiment, for the CR model, another 199/200× of supplied resources were added (1/3 a.u. each for 3 biochemicals out of 30). From this new initial condition, we simulated growth and interaction as described by the mechanistic model for 24 time units, i.e., a digital day. We discarded species with population size <$10^{-5}$ at the end of the 30th cycle and simulated 10 more daily dilution cycles to obtain the equilibrium resident communities. Similarly, we simulated invaders from an initial population of $10^{-3}$ for 10 dilution cycles to get the equilibrium invader populations.

**Simulating invasion dynamics from mechanistic models.** We simulated invasion dynamics from both mechanistic models, similarly to the invasion experiment. 10 wells were initiated with the equilibrium invader population, followed by 10 wells with the equilibrium resident community. We simulated 20 dispersal-dilution-growth cycles. At the beginning of each cycle $n+1$ ($t=0$), we calculated the species composition $\mathbf{N}$ (and resource composition $\mathbf{R}$ for the CR model) for each well position $x$ after dispersal with rate m and dilution with factor $\delta = 1/200$ as:

$$\begin{pmatrix}\mathbf{N}_{x,n+1}\\\mathbf{R}_{x,n+1}\end{pmatrix}_{t=0} = \delta \cdot \left((1-m)\begin{pmatrix}\mathbf{N}_{x,n}\\\mathbf{R}_{x,n}\end{pmatrix}_{t=T} + \frac{m}{2}\begin{pmatrix}\mathbf{N}_{x-1,n}\\\mathbf{R}_{x-1,n}\end{pmatrix}_{t=T} + \frac{m}{2}\begin{pmatrix}\mathbf{N}_{x+1,n}\\\mathbf{R}_{x+1,n}\end{pmatrix}_{t=T}\right) + (1-\delta)\begin{pmatrix}\mathbf{0}_{nS}\\\mathbf{R}_{sup}\end{pmatrix} \tag{7}$$

$\mathbf{0}_{nS}$ is a zero vector with a size equal to the species number, and $\mathbf{R}_{sup}$ is a vector of supplied biochemical concentrations in fresh media. We simulated growth and interaction as described by the mechanistic models for $T = 24$ time units, and used the resulting species and resource compositions ($t=$ T) as the starting point to simulate the next dispersal-dilution-growth cycle.

**Simulating interaction measurements.** Similar to experimental setup, we simulated the interaction measurements by mixing equilibrium invader population and resident community (for CR model also biochemicals) at 29 different fractions ([0.0, 0.0001, 0.001, 0.004, 0.01, 0.02, 0.03, 0.05, 0.1, 0.15, 0.2, 0.25, 0.3, 0.4, 0.5, 0.6, 0.7, 0.75, 0.8, 0.85, 0.9, 0.95, 0.97, 0.98, 0.99, 0.996, 0.999, 0.9999, 1.0], symmetric around 0.5). Then, from these different initial conditions, we simulated the mechanistic model for 1 dilution-growth cycle, specifically, we multiplied the population size (and biochemical concentrations) by $\delta$ and simulated growth and interaction as described by the mechanistic models for $T = 24$ time units. The results were final species compositions for each invader-resident pair and different initial invader fractions.

**Predicting simulated invasion from simulated interaction measurements.** From the initial and final compositions of the simulated interaction measurements described above, we calculated the "volume"-based initial and final fraction of invaders, similar to experimental data. We obtained the interaction curve $g()$ by linearly interpolating the initial and final fractions. All details from the mechanistic model were lost in this process, and we only used the resulting 1-variable interaction curve to predict simulated invasion dynamics following the parameter-free framework in Fig. 3.

**Numerical solver.** We run all simulations in Python version 3.11 with 'solve_ivp' in scipy version 1.13.0. We used the Runge–Kutta–Fehlberg method (RK45) with adaptive stepsize (relTol = 1e-3 and absTol = 1 e-6).

### Reporting summary

Further information on research design is available in the Nature Portfolio Reporting Summary linked to this article.

## Data availability

The experimental and simulation data generated in this study have been deposited on Figshare (https://doi.org/10.6084/m9.figshare.27168834). The experimental data are also available in Supplementary Data 1. The 16S sequencing of resident communities for the multi-strain treatment has been deposited in the European Nucleotide Archive under accession code PRJEB86430.

## Code availability

The code for reproducing all figures and the code for generating the simulation dataset have been archived on Zenodo (https://doi.org/10.5281/zenodo.15274315). The code is also available on Figshare (https://doi.org/10.6084/m9.figshare.27168834).

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

## Acknowledgements

C.R. received funding from the European Research Council (ERC) under the European Union's Horizon 2020 research and innovation programme (grant agreement No 948753), the Deutsche Forschungsgemeinschaft (DFG, German Research Foundation) 468972576 and 540605007, and the Cluster of Excellence EXC 2124 "Controlling Microbes to Fight Infections" (CMFI, EXC 2124 – 390838134). O.S. received funding from the Deutsche Forschungsgemeinschaft 516931136. We thank Hinrich Schulenburg for generously providing the MYb27 strain and Andreas Peschel for generously providing the original *Pseudomonas aeruginosa* PA01 strain. We thank the whole Ratzke lab, Chenlei Hu, and Shuyu He for constructive feedback on the project and the manuscript. The illustrations of bacteria in Figs. 1–3 were made by Bala Akaba (balaakaba@gmail.com).

## Author contributions

X.Y.: Conceptualization, Methodology, Software, Formal analysis, Investigation, Data Curation, Writing - Original Draft, Writing - Review & Editing, Visualization. O.S.: Conceptualization, Methodology, Resources, Writing - Review & Editing. C.R.: Conceptualization, Methodology, Resources, Writing - Original Draft, Writing - Review & Editing, Supervision, Funding acquisition

## Funding

## Competing interests

The authors declare no competing interests.
