## [Transparent Peer Review file · Nature Communications]

Biotic resistance predictably shifts microbial invasion regimes

Corresponding Author: Dr Christoph Ratzke

Version 0:

Reviewer comments:

Reviewer #1

(Remarks to the Author)

The manuscript provides the first experimental demonstration of locked/pulsed invasions and shows the utility of growth rates to predict invasion dynamics. Both of these results are valuable and of broad interest.

I have two major comments.

1. I think the results need better placement in the context of previous work. For example, the Allee effect is not mentioned even once in the text even though that is the key term used in the study of the described phenomena in ecology. More importantly, there has been a lot of theoretical work to investigate pinned and locked invasions and determine the conditions of their onset. I think the only reference to this work is Ref. 51 mentioned in passing on line 314. In my view, it is important to alert the reader to the fact that there is quite a bit of previous work and provide some of the key results. This can easily be done in a few sentences or a paragraph.

2. The theoretical works mentioned above established two important conditions for velocity locking. The first condition includes two factors the patchiness of space and discrete generations. I don't think the latter is mentioned in the manuscript. The second condition is the spatial periodicity. As far as I can tell, all previous studies assumed periodic lattices from the start, so it is not clear if modest spatial variation in dispersal rates, growth rates, or carrying capacities would preserve the locking phenomenon. Given that main motivation behind the work is to develop tools that can be applied to real populations, I think it is worth exploring how important spatial periodicity actually is. These additional simulations would bring the previous theoretical results closer to reality, which I find is the main impetus behind the work.

(Remarks on code availability)

Reviewer #2

(Remarks to the Author)

This study investigates the interplay between dispersal rates and biotic resistance in determining bacterial invasion success in experimental microcosms. Via manipulation of biotic resistance and dispersal rates in both single-population and mixed-population microcosms, the authors identify three distinct invasion regimes that they are able to generally reproduce using a parameter-free modelling framework. This framework does not require prior knowledge on biotic interactions, which is seen as a useful feature in prospects to predict the microbial invasion process in applied contexts.

Major comments:

1- The authors should discuss the issue of temporal autocorrelation of their predictive framework, because their equation is essentially an additive of different parts of the interaction curve – where the state at one part of the curve is dependent on other parts. If this reasoning is correct, therefore, the framework has by necessity a good fit to other data, because the equation is fitted on correlated data of the same experiment. This should be better discussed in the text, and proven otherwise if my reasoning is incorrect.

2- The bidirectionality of dispersal in the experimental design is not clearly justified and begs the question whether this

predictive framework is only useful under this particular experimental design. This bidirectionality leads to a constant homogenization of the communities, which increases the likelihood of obtaining a good fit but compromises generalization. Why did the authors not perform invasions in a directed manner (i.e. without bidirectional transfer), which would actually reflect the dispersal process? As is, the experiment and predictive framework do not describe invasion through dispersal but a slow community homogenization process via repeated mixing of increasingly homogenous communities, a rather odd situation. The authors did find that invasion is in a sense “decomposable over time”, which seems interesting, but is obscured by the stronger claims about the broad applicability of their framework, which are not well supported due to their unusual experimental design. Please clarify to what extent the homogenizing process described here is really a dispersal process, as it currently seems that the applicability of the framework is contingent on the specific, bidirectional, experimental design of this particular study.

3- Previous work (e.g. Jones et al., 2021, eLife <https://doi.org/10.7554/eLife.71811>) showed that community productivity is a stronger driver of invasion success than community composition, but biotic resistance is here manipulated based on taxonomic composition. Please include in the supplementary the absolute OD600 values of the invaded communities, as there is no raw data on the actual growth of the communities.

Based on the comments above, I would not recommend immediate acceptance of this work, as the generality of their findings is unclear and conclusions are only partially supported by the data.

Minor comments:

1. The authors claim that successful invasion depends on dispersal and biotic resistance, but local abiotic conditions in the novel territory are also critical to invader establishment. Of course, abiotic conditions such as pH are often modulated by the local taxa as in this study, but in most environments like soil factors such as water availability, salinity, or particular nutrient levels determine invasion success independently of the local community as long as there is a minimal propagule pressure. Please amend the introduction and discussion as it currently reads as if dispersal and biotic resistance are the only factors involved. This is necessary as the authors hope the predictive framework to be useful in actual ecological invasions beyond microcosms (see e.g. <https://www.nature.com/articles/s41586-023-06378-w>).

2. The authors clearly show that for the mixed community pH does not correlate with biotic resistance, while this is the case in the single-strain system. Do the authors have evidence for which factors might be driving biotic resistance in the mixed community? The discussion is missing some reasoning for why the parameter-free framework might be more useful to predict invasions over actual knowledge of the factors ruling invasion success in a given system, which is information most people in agriculture or gut probiotic design actually have.

3. Did the authors verify that the mixed strain communities remained as such over the experiment? The pulsated invasion regime, which happened at intermediate biotic resistance and dispersal rates, could happen because in those conditions it takes a larger number of transfers to achieve invasion, and more transfers implies more chances for the mixed-strain starting communities to drift into a single-strain situation, and this situation might make invasion possible or impossible at a given point in time depending on how the community drifted, thus leading to the pulsated (on and off) invasion pattern. It would be helpful if the authors could discuss this possibility, and more generally the possible ecological basis of the pulsated invasion regime.

4. Interaction curves and dispersal rates are not so easy to obtain even in highly simplified natural systems. Given the strong impact of pH on invader success in the single strain system, wouldn't it be easier to predict invasion success using information on pH which is usually easier to obtain? This begs some discussion as the predictive framework treats the mixed community as a single unit, which should presumably have a defined impact (such as changing the pH, O₂ levels, or pre-empting certain nutrients in the medium).

Specific comments:

L72. *ureae* in italics.

L96. relax -> reach/achieve/assemble into

L288. Discussions -> Discussion

L479. stabilized -> stabilize; here and throughout the methods section please check spelling errors

L635. Are facilitation and commensalism not included in the consumer-resource model?

Please define “invasion speed” already in the methods section

(Remarks on code availability)

Version 1:

Reviewer comments:

Reviewer #1

(Remarks to the Author)

I am satisfied with all of the changes. I suggest that the manuscript is accepted for publication.

(Remarks on code availability)

Reviewer #2

(Remarks to the Author)

The authors addressed my comments and I only have one last (minor) comment:

The interaction curve is based on 24h growth measurements, a temporal scale that a priori limits the applicability of the presented results to very fast growers. Many bacteria in natural environments have long doubling times, where this short temporal scale would not allow for detecting clear changes in abundance. It would improve the manuscript to briefly discuss, besides practical reasons, why 24h is a suitable scale for calculating the interaction curve, and whether longer times would be more suitable for slower growers or whether this is something worth exploring in the future.

(Remarks on code availability)

We are resubmitting a revised version of our manuscript and thank you for your patience in that regard. We were pleased that our work was recognized as “valuable and of broad interest”. The reviewers raised several points of improvement: (1) testing the generality of our modeling framework, in particular concerning spatial heterogeneity (reviewer 1) and different dispersal schemes (reviewer 2), (2) connecting the findings better to existing theoretical work (reviewer 1) and discussing the biological meaning of our results more thoroughly (reviewer 2), and (3) explaining and justifying the experimental design in more depth (reviewer 2).

These and further points have all been addressed in the response letter. We especially explored point (1) with a variety of additional simulations that show that the general features and the predictive power of our model system hold over a wide range of conditions, which is underlined with 4 additional supplementary figures. For point (2), we updated the introduction, results, discussion, and two supplementary figure panels to more thoroughly discuss previous work and explain the biological basis of our results. For point (3), we updated results and one main figure panel to better explain our experimental setup. Based on the reviewers' suggestions we also improved the manuscript in many other ways as laid out in the point-by-point response. We believe that these changes substantially improved the manuscript and would like to thank the reviewers for their very valuable suggestions.

We attach a point-by-point response and a PDF file highlighting the changes. Thank you for your attention to our manuscript.

We look forward to hearing from you,
Xiaozhou Ye
Or Shalev
Christoph Ratzke

Reviewer #1 (Remarks to the Author):

The manuscript provides the first experimental demonstration of locked/pulsed invasions and shows the utility of growth rates to predict invasion dynamics. Both of these results are valuable and of broad interest.

Thanks for your recognition of the novelty and importance of our work!

I have two major comments.

1. I think the results need better placement in the context of previous work. For example, the Allee effect is not mentioned even once in the text even though that is the key term used in the study of the described phenomena in ecology. More importantly, there has been a lot of

theoretical work to investigate pinned and locked invasions and determine the conditions of their onset. I think the only reference to this work is Ref. 51 mentioned in passing on line 314. In my view, it is important to alert the reader to the fact that there is quite a bit of previous work and provide some of the key results. This can easily be done in a few sentences or a paragraph.

Thanks for the suggestion! We extended our discussion about previous theoretical work accordingly, as below:

Lines 323-333: “Theoretical studies modeled invasion as a single species reaction-diffusion process and identified two conditions for pulsed and pinned waves. The first condition is patchy instead of continuous habitats(Keitt et al., 2001; Mitkov et al., 1998; Wang et al., 2019), as is the case for our experiment and many natural scenarios(Leibold et al., 2004). The second condition is Allee effect(Lewis and Kareiva, 1993; Taylor and Hastings, 2005; Wang et al., 2019, 2002), which means the growth rate of a species reduces when the population size drops, often due to the need for cooperative growth(Allee, 1938). In our study, biotic resistance of resident communities makes it harder for the invader to grow at low population density as compared to high density, leading to similar dynamics as caused by an Allee effect. Although we performed dispersal as a discrete daily event, pinned and pulsed invasion can also happen under continuous dispersal(Mitkov et al., 1998; Wang et al., 2019). Our results indicate that simplified theories about the spread of a single species can be applied to more complex scenarios where resident communities are involved.”

2. The theoretical works mentioned above established two important conditions for velocity locking. The first condition includes two factors the patchiness of space and discrete generations. I don't think the latter is mentioned in the manuscript.

Thanks for pointing this out, we will discuss the two preconditions in the following. First, we would like to distinguish between pulsed and locked waves. Pulsed waves advance in “bursts” separated by seemingly stationary periods, whereas locked waves take only rational values as invasion speed and advance with strict periodicity. (1) Pulsed waves do not require discrete generation times, whereas locked waves do. (2) Pulsed waves may not be velocity locked; what we observe experimentally are surely pulsed waves, but we cannot confirm that they are also locked waves. Therefore, to avoid confusing the reader with these different terminologies, we would kindly prefer to not discuss locked wave in the manuscript. We discuss points (1) and (2) in more depth in the following.

(1) Pulsed waves do not require discrete generations

Previous studies of reaction-diffusion equations have shown that discrete generation is not a precondition for pulsed waves (Mitkov et al., 1998). To further explore this for our system, we simulated 8 examples - most are pulsed - under both discrete and continuous generations. Indeed, we observed pulsed invasion also under continuous generation time (Fig. Rev1, especially in examples 1, 7, 8).

Fig.Rev1 Pulsed invasion can also happen in continuous generation. Each row is an example with the given resident community (rsd), invader (ivd) and dispersal rate (m), all examples except example 6 are pulsed. On the left most column are results for discrete generation and on the right continuous generation. For each simulation, we showed instantaneous invasion speed over

time (blue curve) as well as invader fraction across spatial patches over time (grey heatmap). We simulated continuous generation by splitting 1 day into 100 cycles of interaction and dispersal, so that the generation time (1/100 day) is roughly at the same time scale of species interactions. We additionally simulated example 1 under 1/1000 day generation time (the right most panel) to make sure the dynamics stay the same as the generation time further approaches the continuous limit. For each cycle, we recalculated the per cycle dilution as $d_N = d \cdot (1/N)$ where d is the daily dilution rate and N is the number of cycles per day. We recalculated the dispersal rate as $m_N = m / N$ where m is the daily dispersal rate, however we do not expect the invasion rate to reflect the same invader pressure for discrete- and continuous- generation scenario, because dispersal happens at different stages of invader growth.

(2) In our system we observe pulsed but not necessarily locked waves

Pulsed wave is different from velocity locked wave, where for the latter invasion speed can only take rational values. It is true that the invasion dynamics in velocity locked waves are pulsed given that the average invasion speed is low, but pulsed waves can happen without being velocity locked. To demonstrate this, we simulated the invasion of an example invader into an example resident community, changing dispersal rate m at a very fine scale. We additionally simulated the invasion as a single species growth-dispersal process, where the growth function takes the shape of the interaction curve obtained from the multi-species simulation. As could be seen from a region of example dynamics (Fig. Rev2), all invasions in this region are pulsed, but only some (highlighted in orange) are velocity locked as evident from their periodicity. Therefore, simulations suggest that – even theoretically – velocity may not be locked in the pulsed waves that we observed.

Furthermore, distinguishing locked waves from general pulsed waves in our experimental data is very challenging. Because the experimental noise makes it very difficult to judge if the observed velocity fluctuations are strictly periodic and whether the invasion speed stays the same over a range of dispersal rates. Thus, despite some of the pulsed waves we observed might be locked, confirming this hypothesis would require significantly eliminating noise and measuring invasion speed under many more dispersal rates, which are not experimentally feasible given the large scale of these experiments.

Fig.Rev2 Examples of pulsed invasion waves in single-species and in multi-species simulations, highlighting that pulsed wave may not be velocity locked. Daily invasion speed as dispersal rate m gradually increases from 0.080 to 0.091, horizontal dashed lines represent the average invasion speed across days. Velocity locked waves are highlighted in orange; all others are pulsed but not locked waves. To compare better with previous theoretical results, when computing invasion speed we defined front position as the sum of invader fraction divided by carrying capacity (equals 1) instead of position above the 50% invader threshold; the conclusions stayed the same regardless. Invasion was simulated in an example where an invader (ivd4) invades into a resident community (rsd0) with 5 species. The invasion was simulated on 50 patches for 200 dispersal rates and 1000 days, with an initial distribution of 10 invader patches followed by 40 resident patches. For the multi-species simulation, we simulated species interactions following the general Lotka-Volterra model. For the single-species simulation, we simulated species growth with a one-dimensional map $N_{n+1}=g(N_n)$ from population size at day n (N_n) to day $n+1$ (N_{n+1}), where $g()$ has the same shape as the interaction curve from multi-species simulation (i.e. equivalent to what we do for prediction framework).

While not directly related to the discussions above but maybe of interest for the reviewer, we also noticed that velocity locked waves occur less often in the multi-species simulation as compared to the single-species invasion simulation. This could be confirmed by the invasion speed – dispersal rate relationship in Fig. Rev3, where the relationship from single-species simulation has more visible velocity locked “steps” as comparing to multi-species simulation. This indicates that when multiple species and more complex dynamics are involved – as the case in our study - velocity locking might become less common, which can be an interesting topic for future studies.

Fig.Rev3 Velocity locking is less frequent in multi-species than in single-species simulations. Mean invasion speed v as a function of dispersal rate m , where the steps caused by velocity locking could be observed. Same simulation procedure as in Fig. Rev1.

Taken together, the pulsed invasion we observed is not necessarily velocity locked and does not require discrete generation to occur.

The second condition is the spatial periodicity. As far as I can tell, all previous studies assumed periodic lattices from the start, so it is not clear if modest spatial variation in dispersal rates, growth rates, or carrying capacities would preserve the locking phenomenon. Given that main motivation behind the work is to develop tools that can be applied to real populations, I think it is worth exploring how important spatial periodicity actually is. These additional simulations would bring the previous theoretical results closer to reality, which I find is the main impetus behind the work.

This is a very good point! The effects of spatial heterogeneity on invasion dynamics are indeed hardly addressed. For velocity locking, the most related results are from Wang et al., 2019, where the authors found that demographic noise, spatial fluctuation of growth rates, and temporal fluctuation of dispersal rates do not impact locking. For pulsed invasion in

general, to investigate the impact of heterogeneous spatial patches we simulated invasion of 8 examples (the same 8 as used in Fig. Rev1), with increasing heterogeneity in dispersal rate, carrying capacity, and growth rates respectively.

The figures below show the daily invasion speed from these simulations. Compared to the scenario without heterogeneity ($H=0$), increasing heterogeneity in dispersal rate m (Fig. S17) or carrying capacity K (Fig. S18) leads to increasingly irregular intervals between pulses, but the overall invasion dynamics remains similar and pulsed. Increasing heterogeneity in growth rates across patches has a higher impact on invasion dynamics (Fig. S19), the invasion is still pulsed, but the pulses are less regular and can be easily stuck / accelerated. Interestingly, despite relatively large changes of daily invasion dynamics, the mean invasion speed stays similar even under very high spatial heterogeneity in growth rate.

We added these results as supplementary figures (Fig. S17-S19), and mentioned the conclusions in main text as below:

Lines 340-343: “Pulsed waves are only mildly affected by spatial heterogeneity in dispersal rate (Supplementary Fig. 17) and carrying capacity (Supplementary Fig. 18); heterogeneity in growth rates has a bigger impact and can frequently lead to irregular pulses (Supplementary Fig. 19).”

Fig.S17 Daily invasion speed for invasion along patches with heterogeneous dispersal rate m . Each row shows daily invasion speed (solid blue line) and mean invasion speed (horizontal blue dash) for an example with the given resident community (rsd), invader (ivd) and dispersal rate (m). The heterogeneity of dispersal rate m across patches increases from left to right. For given heterogeneity level H , the dispersal rate m out of each patch is multiplied by a random value independently drawn from the uniform distribution $[1-H, 1+H]$, whereas $H=0$ means no heterogeneity. To compare better with previous theoretical results, we defined the front position as the sum of invader fractions (divided by carrying capacity 1) when computing the invasion speed instead of the position above the 50% threshold; the conclusions stayed the same regardless.

Fig.S18 Daily invasion speed for invasion along patches with heterogeneous carrying capacity K. Each row shows daily invasion speed (solid blue line) and mean invasion speed (horizontal blue dash) for an example with the given resident community (rsd), invader (ivd) and dispersal rate (m). The heterogeneity of carrying capacity K across patches increases from left to right. For given heterogeneity level H, the carrying capacity of each patch is independently drawn from the uniform distribution $[1-H, 1+H]$, $H=0$ means no heterogeneity. Since we normalized carrying capacity in Lotka-Volterra model to 1, this is equivalent to divide the interaction terms α by these randomly drawn K. To compare better with previous theoretical results, we defined the front position as the sum of invader fractions (divided by carrying capacity 1) when computing

the invasion speed instead of the position above the 50% threshold; the conclusions stayed the same regardless.

Fig.S19 Daily invasion speed for invasion along patches with heterogeneous growth rate r . Each row shows daily invasion speed (solid blue line) and mean invasion speed (horizontal blue dash) for an example with the given resident community (rsd), invader (ivd) and dispersal rate (m). The heterogeneity of growth rates r across patches increases from left to right. For given heterogeneity level H , the growth rate r for each species in each patch is multiplied by a random value drawn independently from the uniform distribution $[1-H, 1+H]$, $H=0$ means no heterogeneity. To compare better with previous theoretical results, we defined the front position as the sum of invader fractions (divided by carrying capacity 1) when computing the invasion

speed instead of the position above the 50% threshold; the conclusions stayed the same regardless.

Reviewer #2 (Remarks to the Author):

This study investigates the interplay between dispersal rates and biotic resistance in determining bacterial invasion success in experimental microcosms. Via manipulation of biotic resistance and dispersal rates in both single-population and mixed-population microcosms, the authors identify three distinct invasion regimes that they are able to generally reproduce using a parameter-free modelling framework. This framework does not require prior knowledge on biotic interactions, which is seen as a useful feature in prospects to predict the microbial invasion process in applied contexts.

Thanks for thoroughly reading our manuscript!

Major comments:

1- The authors should discuss the issue of temporal autocorrelation of their predictive framework, because their equation is essentially an additive of different parts of the interaction curve – where the state at one part of the curve is dependent on other parts. If this reasoning is correct, therefore, the framework has by necessity a good fit to other data, because the equation is fitted on correlated data of the same experiment. This should be better discussed in the text, and proven otherwise if my reasoning is incorrect.

Thanks for bringing up this important concern. We should indeed clarify the relationship between the interaction curve and the invasion prediction better! The different points of the interaction curve are actually independent, because they were measured in independent measurements, e.g., from separate bacterial cultures. Specifically, we obtained the bacteria cultures by mixing the invader and the resident community in different initial fractions, then we measured their final invader fractions after 24 hours. Since different cultures were physically separate from each other and were measured after the same time period, there was no temporal sequence nor dependency between different parts of the interaction curve. It is true that the interaction curve is often continuous and monotonic, but this is due to biological properties of interactions, rather than temporal correlation as in time-series data.

We additionally would like to emphasize that the interaction curves were not obtained from the invasion experiments but measured in separate interaction experiments. This is highlighted by the fact that the same interaction curve could be used to predict the outcome of different invasion experiments with different dispersal rates (as shown in main text Fig. 3).

To extend the above discussions, we would like to further share our insights regarding the prediction framework. One of the key motivations behind the prediction framework is to see how well we could approximate the ever-changing interactions between species during an invasion with the interaction curve. There are two major benefits of measuring the interaction curve as compared to the invasion experiment. First, invasion must be observed over extended time period and space, but interaction curve can be measured over a short period and is therefore much easier to obtain. Second, an invasion involves both dispersal and interaction processes, while the interaction curve segregates out only the latter and can be applied to different dispersal scenarios.

We updated main text Fig. 3a to better demonstrate how interaction curves were measured:

Fig.3a Measurement of interaction curve. The interaction curve is obtained by mixing the invader and the resident community in different mixing ratios and measuring the invader fraction 24 hours later.

We extended the main text to better clarify these points, as below:

Lines 206-208: “The interaction curve is obtained by mixing the invader and the resident community in different mixing ratios and measuring the invader fraction 24h later.”

Lines 226-229: “It is important to note that we measured the interaction curves in a separate experiment and did not obtain them from the invasion data. Furthermore, the points on the interaction curve are not correlated with each other as we measured them from independent measurements, i.e. different bacterial cultures.”

Lines 248-251: “Measuring an interaction curve is much faster and easier than tracking an invasion over time, allowing us to predict invasions before they even occur. Moreover, since the interaction curve isolates the interaction component from the invasion process, it can be applied to different dispersal rates and invasion scenarios.”

2- The bidirectionality of dispersal in the experimental design is not clearly justified and begs the question whether this predictive framework is only useful under this particular experimental design. This bidirectionality leads to a constant homogenization of the communities, which increases the likelihood of obtaining a good fit but compromises generalization. Why did the authors not perform invasions in a directed manner (i.e. without bidirectional transfer), which would actually reflect the dispersal process? As is, the experiment and predictive framework do not describe invasion through dispersal but a slow community homogenization process via repeated mixing of increasingly homogenous communities, a rather odd situation. The authors did find that invasion is in a sense “decomposable over time”, which seems interesting, but is obscured by the stronger claims about the broad applicability of their framework, which are not well supported due to their unusual experimental design. Please clarify to what extent the homogenizing process described here is really a dispersal process, as it currently seems that the applicability of the framework is contingent on the specific, bidirectional, experimental design of this particular study.

Thanks for raising this important question! Indeed, the accuracy of our prediction framework does not depend on whether the dispersal is bidirectional or unidirectional. This is because the framework's core is to approximate complex interactions by the interaction curve, which is dispersal independent. We demonstrate this by simulating unidirectional dispersal with the general Lotka-Volterra model and comparing the simulations to the predictions from the corresponding interaction curve. The results are shown in Fig. S13 below, the upper panels show the distribution of prediction error, and the lower panels show the predicted vs. simulated invasion speed for different invasion scenarios. Especially for forward invasion, which we care about most, the prediction accuracy remains high for unidirectional dispersal.

Fig.S13 Prediction is highly accurate also when dispersal is unidirectional. Predicted vs. simulated invasion speed (v-viii) and the distributions of prediction errors (i-iv) for generalized Lotka-Volterra model and unidirectional dispersal. In addition, predictions for the three distinct scenarios are shown separately: (1) forward invasion (i, v, orange), where invader takes over and the invasion front moves forward; (2) failed invasion (ii, vi, blue), where the invader could not grow within the resident community and the invasion front does not move; and (3) partial invasion (iii, vii, purple), where the invader coexists with the resident community, since the invader cannot exclude the residents, only part of the invasion front moves forward. These scenarios are analogous to forward, backward including pinned, and bidirectional invasions when dispersal is bidirectional as shown in Supplementary Figs. 11-12.

We focused on bidirectional dispersal because dispersal, contrary to animal migration, is generally described as a diffusion process that happens equally likely in all possible directions, as can be seen from existing literature (Clobert, 2012). This should be especially relevant for microbes as they normally disperse passively with wind, water, human action, etc., which is on average not directed. Dispersal, whether multidirectional or unidirectional, is indeed a homogenizing force, but due to species interactions it is not just averaging out communities in the sense that there is no longer distinction between the invader and the resident communities. Take as an example when an invader is in contact with resident community without any resistance: although the invader and resident species mix into each

other by dispersal, the resulting invasion is still unidirectional, because the invader that disperses into resident species takes over but the resident species that disperse into the invader population die out.

We added Fig. S13 to the supplement, and mentioned it in main text as below:

Lines 281-283: “The framework can also predict invasion in other dispersal scenarios, for instance when dispersal is unidirectional instead of bidirectional (Supplementary Fig. 13).”

3- Previous work (e.g. Jones et al., 2021, eLife <https://doi.org/10.7554/eLife.71811>) showed that community productivity is a stronger driver of invasion success than community composition, but biotic resistance is here manipulated based on taxonomic composition. Please include in the supplementary the absolute OD600 values of the invaded communities, as there is no raw data on the actual growth of the communities.

This is a good point, thanks for the suggestion. We added the OD600nm data to Supplementary Fig. 2 as panel d, which is also shown here. As can be seen, as resistance increases from Comm A to Comm D, the absolute OD600 of the invaded community does not show any trend of increasing, meaning that productivity is not the major driver of biotic resistance in our system.

Fig.S2d Biotic resistance of resident community did not correlate with productivity. The productivity of resident communities was measured by OD600nm on day 1 of the invasion experiment. As biotic resistance increased from Comm A to Comm D, productivity did not show a similar increasing trend.

We also modified main text to include this information as below:

Lines 101-102: “Different from previous studies(Jones et al., 2021), the community productivity (OD_{600nm}, Supplementary Fig. 2d) also does not determine biotic resistance in our systems.”

Based on the comments above, I would not recommend immediate acceptance of this work, as the generality of their findings is unclear and conclusions are only partially supported by the data.

Minor comments:

1. The authors claim that successful invasion depends on dispersal and biotic resistance, but local abiotic conditions in the novel territory are also critical to invader establishment. Of course, abiotic conditions such as pH are often modulated by the local taxa as in this study, but in most environments like soil factors such as water availability, salinity, or particular nutrient levels determine invasion success independently of the local community as long as there is a minimal propagule pressure. Please amend the introduction and discussion as it currently reads as if dispersal and biotic resistance are the only factors involved. This is necessary as the authors hope the predictive framework to be useful in actual ecological invasions beyond microcosms (see e.g. <https://www.nature.com/articles/s41586-023-06378-w>).

Thanks for the suggestion, abiotic condition surely also plays a very important role. Indeed, the absolute resistance we measured is a combined result of biotic and abiotic factors, as we could imagine if the invader cannot survive in the media at all, the resistance would be very high regardless of resident community. We amended introduction and discussion accordingly:

Lines 33-37: “The dispersing microbes only thrive under the right nutrients(Schäfer et al., 2023), pH(Dumbrell et al., 2010), temperature(Garcia-Pichel et al., 2013) or salinity conditions(Oren, 2008). Accordingly, abiotic environmental factors of the novel territory can decide about the establishment of an invading species. However, sequencing data of environmental samples show that microbe species are absent from many habitats that they in principle could live in(Fierer and Jackson, 2006; Martiny et al., 2006).”

Lines 184-186: “The interaction curve depends on both the interacting species and how these interactions are impacted by the abiotic environment, and thus summarizes both the biotic and abiotic resistance.”

Lines 311-314: “Consequently, the invasion outcome not only depends on how fast an invader can spread into a novel territory and the abiotic conditions of the new habitat but also on how it interacts with the resident organisms at the new location.”

2. The authors clearly show that for the mixed community pH does not correlate with biotic resistance, while this is the case in the single-strain system. Do the authors have evidence for which factors might be driving biotic resistance in the mixed community? The discussion is missing some reasoning for why the parameter-free framework might be more useful to predict invasions over actual knowledge of the factors ruling invasion success in a given system, which is information most people in agriculture or gut probiotic design actually have.

We do speculate that biotic resistance in mixed community might be driven by niche similarity. We don't have direct evidence, but the most resistant community is made of *Pseudomonas* strains, i.e. from the same family as our invader *Pseudomonas aeruginosa*. We added this information to discussion as below.

Lines 102-105: “Interestingly, the most resistant community (Comm D) consists of *Pseudomonas* strains (Supplementary Fig. 2b), i.e. closely related to the invader, suggesting that niche similarity might play a role in determining resistance levels.”

Actual knowledge of the system is indeed very important; however, it can normally give only qualitative and comparative understandings, but not quantitative predictions. For instance, although we know that in the two-strain system the resident species inhibits invader by reducing pH, we cannot tell directly how much this would slow down invasion and whether the pinned wave can occur. Such predictions would require knowing all quantitative details of the interactions between the invader and the residents, which can be quite complicated and depend on the population density of the species, thus are generally hard to obtain. Indeed, predicting dynamics of microbial communities bottom up from molecular mechanisms is a major - and currently mostly unresolved - challenge in microbial ecology (Widder et al., 2016). The interaction curve in this study not only circumvents this challenge to reach meaningful predictions, but also provides a way to link actual knowledge to invasion dynamics. It is much easier to examine the impacts of different (a)biotic factors on the interaction curve, then apply the prediction framework to further infer their impacts on the invasion outcome.

3. Did the authors verify that the mixed strain communities remained as such over the experiment? The pulsated invasion regime, which happened at intermediate biotic resistance and dispersal rates, could happen because in those conditions it takes a larger number of transfers to achieve invasion, and more transfers implies more chances for the mixed-strain starting communities to drift into a single-strain situation, and this situation

might make invasion possible or impossible at a given point in time depending on how the community drifted, thus leading to the pulsated (on and off) invasion pattern. It would be helpful if the authors could discuss this possibility, and more generally the possible ecological basis of the pulsated invasion regime.

Yes, we did verify that mixed strain community retained its species composition over the course of the experiment. We cultured the resident communities for 4 days and verified that it remained stable over this period (Supplementary Fig. 2b, updated version below). Additionally, on day 8 of the 10-day invasion experiment, we plated the communities on agar plates to assess composition by colony forming units (CFU). Although we did not count the obtained CFUs at that time, we captured microscopic images of the agar plates. We re-analyzed these images to obtain a rough estimation of resident community composition on day 8. As seen from updated Supplementary Figure S2b below (Fig. S2b, “inv8”), even at this late stage of invasion experiment the resident communities still consisted of mixed strains as in the precultures.

Fig.S2b Compositions of 4 resident communities. Resident community compositions over 4 days of 1/200X daily dilution after reviving from glycerol stock (days 0-4), for 3 replicates each initiated from an independent inoculum. For each resident community, compositions of 3 wells on day 8 of the invasion experiment (inv8) are also shown. The 3 wells from invasion experiment were taken from the 3 lowest dispersal rates and the farthest from the invasion front, thus least influenced by invader – they only encountered the invader for 1 day. Species compositions were obtained by counting colonies with distinct morphologies (CFU) and additionally by 16S sequencing on day 4.

Besides the direct observation above, there are a few additional reasons that the pulsed invasion regime was unlikely the result of random species loss: (1) The pulsed invasion is also observed in the two-strain system, where the resident “community” consists of a single strain, so random species loss could not happen, (2) the presence of dispersal makes random extinction unlikely, as extinct population can always be rescued by new individuals from nearby wells.

The ecological basis of the pulsed invasion is that with biotic resistance the invader population can only grow slowly at the beginning, but as the invader builds up and the resident populations decline, the resistance weakens, and the invader breaks through. This has been shown theoretically before (Keitt et al., 2001) and is also supported by tracking invader CFU daily as shown in Supplementary Fig. 5f and attached below. Take well 4 as an example, the invasion happened suddenly on day 4 when observed on the linear scale (upper panel), but when observed on the log10 scale (lower panel) the invader population had been steadily increasing over time.

Fig.S5 Dynamics of the absolute invader (S_u) population in pinned and pulsed invasions. Each panel shows results from a different dispersal rate for the two-strain system with 40mM phosphate buffer (intermediate resistance). **(f)** In pulsed invasion, the invader slowly accumulated at the invasion front before suddenly taking over. Upper left: daily and mean invasion speeds; lower left: invasion waves; both are as described in Main Fig. 2b. Right: daily absolute invader population size measured by CFU, presented in both linear (upper) and log10 (lower) scales, from the first well being invaded (well 4) to the last well with a detectable invader population.

We updated Supplementary Fig. 2b and extended our discussions in main text as below:

Lines 141-147: “In pulsed invasion, the invasion front advances in bursts (fast invasion phases) separated by seemingly stationary periods (slow invasion phases) (Fig. 2c, II). During the slow invasion phase, the small invader population faces relatively strong biotic resistance and accumulates slowly at the invasion front

(Supplementary Fig. 5e-g); during the fast invasion phase, the invader population growth is large enough to break through the biotic resistance and rapidly take over a new habitat.”

4. Interaction curves and dispersal rates are not so easy to obtain even in highly simplified natural systems. Given the strong impact of pH on invader success in the single strain system, wouldn't it be easier to predict invasion success using information on pH which is usually easier to obtain? This begs some discussion as the predictive framework treats the mixed community as a single unit, which should presumably have a defined impact (such as changing the pH, O₂ levels, or pre-empting certain nutrients in the medium).

It is true that measuring pH itself is an easy task and may give us some hints about invasion success - given the prior knowledge that pH plays an important role. However, to be able to predict invasion we not only need to measure how much the invader and resident species change pH, but also how different pH impacts the growth rates of invader and resident species. This information can be even more difficult to measure than the interaction curve.

More importantly, pH may not be that important for many invasion processes, just like the case of the multi-strain system. The effect of the resident community on the invader is in general a combined effect of many different interaction mechanisms, including pH, nutrient competition, toxin production, etc., with their relative importance vary for different systems. Despite treating the resident community as a single unit, it is still difficult to measure how the resident community changes all these factors and to what extent such changes impact the invader. In fact, one important line of research in our lab is to investigate the drivers of interactions, and our preliminary results from mass spectrometry and genomic show that interactions are driven by a large number of factors. It is even more challenging to integrate such information into a combined effect to predict invasion outcome. Therefore, it remains an unsolved scientific challenge to predict invasion based on how resident communities change the environment directly and our approach offers a way to overcome this unsolved scientific problem.

Specific comments:

L72. *ureae* in italics.

Corrected.

L96. relax -> reach/achieve/assemble into

We modified the sentence to: "... mixing various sets of strains and letting them ~~relax to settle into~~ their stable states ..."

L288. Discussions -> Discussion

Corrected.

L479. stabilized -> stabilize; here and throughout the methods section please check spelling errors

Thanks! We worked through the methods section again and corrected spelling and grammar errors.

L635. Are facilitation and commensalism not included in the consumer-resource model?

Facilitation and commensalism are included also in the consumer-resource model. These positive interactions can happen in the form of cross feeding - when one species secretes metabolic byproducts that could be used as resource by another species; or in the form of cross protection – when one species utilizes a resource which is toxic for another species.

Please define "invasion speed" already in the methods section

Thanks for the suggestion, we added the definition to the methods section as below:

Lines 598-601: "We defined daily invasion speed on day n (v_n) as the distance (in wells) the invasion front moved forward on that day, and calculated the mean invasion speed v as the average daily invasion speed from day 3 until the end of the experiment or until the invader took over all wells."

References:

Allee, W.C., 1938. The Social Life of Animals. Heinemann.

Clobert, J. (Ed.), 2012. Dispersal ecology and evolution, 1st ed. ed. Oxford University Press, Oxford.

Dumbrell, A.J., Nelson, M., Helgason, T., Dytham, C., Fitter, A.H., 2010. Relative roles of niche and neutral processes in structuring a soil microbial community. ISME J. 4, 337–345. <https://doi.org/10.1038/ismej.2009.122>

Fierer, N., Jackson, R.B., 2006. The diversity and biogeography of soil bacterial communities. Proc. Natl. Acad. Sci. 103, 626–631. <https://doi.org/10.1073/pnas.0507535103>

- Garcia-Pichel, F., Loza, V., Marusenko, Y., Mateo, P., Potrafka, R.M., 2013. Temperature Drives the Continental-Scale Distribution of Key Microbes in Topsoil Communities. *Science* 340, 1574–1577. <https://doi.org/10.1126/science.1236404>
- Jones, M.L., Rivett, D.W., Pascual-García, A., Bell, T., 2021. Relationships between community composition, productivity and invasion resistance in semi-natural bacterial microcosms. *eLife* 10, e71811. <https://doi.org/10.7554/eLife.71811>
- Keitt, T.H., Lewis, M.A., Holt, R.D., 2001. Allee Effects, Invasion Pinning, and Species' Borders. *Am. Nat.* 157, 203–216. <https://doi.org/10.1086/318633>
- Leibold, M.A., Holyoak, M., Mouquet, N., Amarasekare, P., Chase, J.M., Hoopes, M.F., Holt, R.D., Shurin, J.B., Law, R., Tilman, D., Loreau, M., Gonzalez, A., 2004. The metacommunity concept: a framework for multi-scale community ecology. *Ecol. Lett.* 7, 601–613. <https://doi.org/10.1111/j.1461-0248.2004.00608.x>
- Lewis, M.A., Kareiva, P., 1993. Allee Dynamics and the Spread of Invading Organisms. *Theor. Popul. Biol.* 43, 141–158. <https://doi.org/10.1006/tpbi.1993.1007>
- Martiny, J.B.H., Bohannan, B.J.M., Brown, J.H., Colwell, R.K., Fuhrman, J.A., Green, J.L., Horner-Devine, M.C., Kane, M., Krumins, J.A., Kuske, C.R., Morin, P.J., Naeem, S., Øvreås, L., Reysenbach, A.-L., Smith, V.H., Staley, J.T., 2006. Microbial biogeography: putting microorganisms on the map. *Nat. Rev. Microbiol.* 4, 102–112. <https://doi.org/10.1038/nrmicro1341>
- Mitkov, I., Kladko, K., Pearson, J.E., 1998. Tunable Pinning of Burst Waves in Extended Systems with Discrete Sources. *Phys. Rev. Lett.* 81, 5453–5456. <https://doi.org/10.1103/PhysRevLett.81.5453>
- Oren, A., 2008. Microbial life at high salt concentrations: phylogenetic and metabolic diversity. *Saline Syst.* 4, 2. <https://doi.org/10.1186/1746-1448-4-2>
- Schäfer, M., Pacheco, A.R., Künzler, R., Bortfeld-Miller, M., Field, C.M., Vayena, E., Hatzimanikatis, V., Vorholt, J.A., 2023. Metabolic interaction models recapitulate leaf microbiota ecology. *Science* 381, eadf5121. <https://doi.org/10.1126/science.adf5121>
- Taylor, C.M., Hastings, A., 2005. Allee effects in biological invasions. *Ecol. Lett.* 8, 895–908. <https://doi.org/10.1111/j.1461-0248.2005.00787.x>
- Wang, C.-H., Matin, S., George, A.B., Korolev, K., 2019. Pinned, locked, pushed, and pulled traveling waves in structured environments. *Theor. Popul. Biol.* <https://doi.org/10.1016/j.tpb.2019.04.003>
- Wang, M.-H., Kot, M., Neubert, M.G., 2002. Integrodifference equations, Allee effects, and invasions. *J. Math. Biol.* 44, 150–168. <https://doi.org/10.1007/s002850100116>
- Widder, S., Allen, R.J., Pfeiffer, T., Curtis, T.P., Wiuf, C., Sloan, W.T., Cordero, O.X., Brown, S.P., Momeni, B., Shou, W., Kettle, H., Flint, H.J., Haas, A.F., Laroche, B., Kreft, J.-U., Rainey, P.B., Freilich, S., Schuster, S., Milferstedt, K., van der Meer, J.R., Großkopf, T., Huisman, J., Free, A., Picioreanu, C., Quince, C., Klapper, I., Labarthe, S., Smets, B.F., Wang, H., Isaac Newton Institute Fellows, Soyer, O.S., 2016. Challenges in microbial ecology: building predictive understanding of community function and dynamics. *ISME J.* 10, 2557–2568. <https://doi.org/10.1038/ismej.2016.45>

Reviewer #2 (Remarks to the Author):

The authors addressed my comments and I only have one last (minor) comment:

The interaction curve is based on 24h growth measurements, a temporal scale that a priori limits the applicability of the presented results to very fast growers. Many bacteria in natural environments have long doubling times, where this short temporal scale would not allow for detecting clear changes in abundance. It would improve the manuscript to briefly discuss, besides practical reasons, why 24h is a suitable scale for calculating the interaction curve, and whether longer times would be more suitable for slower growers or whether this is something worth exploring in the future.

Thanks for the suggestion. It is true that the prediction accuracy relies to some extent on the temporal scale of measuring the interaction curve. What matters most though is not how fast the species grow, but that the prediction is more accurate if the time scale for interaction measurement is equal to or smaller than the typical time scale of dispersal (i.e. the time interval between dispersal events). If the interaction measurements are of similar time scale as the dispersal, we can use them directly for prediction; if the time scale is smaller, we can “integrate” it over time to get the interaction curve at the desired time scale; however, if the time scale gets longer, some fine details of species interaction might not be captured, and prediction accuracy would gradually decrease.

We clarified this point in the main text:

"Also, for optimal prediction accuracy, the species interactions should be measured at a similar or shorter time scale than the occurrence of the dispersal events (see Supplementary Note 2 for clarification), which can become experimentally challenging in case of frequent dispersals."

and as Supplementary Note 2:

“To apply the prediction framework to a new system, it is important to ensure that the time scales of interaction measurements and dispersal match. This involves two aspects:

First, the time units in the equation in Fig. 3a should be consistent. For instance, in our study, the interaction curve describes the change of the invader fraction after 24 hours, and the dispersal rate describes how much dispersal occurs every 24 hours. This does not require that dispersal must occur exactly at the same time

interval as the interaction measurements, but that these two should be normalized to the same unit.

Second, to achieve the highest accuracy, it is advisable that the time scale for interaction measurement be equal to or smaller than the typical time scale of dispersal (i.e. the time interval between dispersal events). If the interaction measurements are on a similar time scale as the dispersal, we can use the equation in Fig.3a directly for prediction. If the time scale is smaller, we can “integrate” the interactions over time to get the interaction curve at the desired time scale. However, if the time scale is bigger, some fine details of species interaction might not be captured as the interaction time scale increases, and prediction accuracy would gradually decrease.

We note that the second aspect is not a strict requirement but would help improve prediction accuracy.”